# PYRREGULAR: A UNIFIED FRAMEWORK FOR IRREGULAR TIME SERIES, WITH CLASSIFICATION BENCHMARKS

**Francesco Spinnato**[1], **Cristiano Landi**[2]

University of Pisa, Pisa, Italy · [1]`francesco.spinnato@unipi.it` [2]`cristiano.landi@phd.unipi.it`

## ABSTRACT

Irregular temporal data, characterized by varying recording frequencies, differing observation durations, and missing values, presents significant challenges across fields like mobility, healthcare, and environmental science. Existing research communities often overlook or address these challenges in isolation, leading to fragmented tools and methods. To bridge this gap, we introduce a unified framework, and the first standardized dataset repository for irregular time series classification, built on a common array format to enhance interoperability. This repository comprises 34 datasets on which we benchmark 12 classifier models from diverse domains and communities. This work aims to centralize research efforts and enable a more robust evaluation of irregular temporal data analysis methods.

## 1 INTRODUCTION

High-dimensional temporal data is increasingly accessible to decision-makers, domain experts, and researchers (Shumway et al., 2000). It is vital in fields like mobility, healthcare, and environmental science to capture dynamic changes over time. Yet, variations in recording frequencies, durations across sensors, and occasional failures lead to signals with unequal lengths, gaps, and missing values (Harvey et al., 1998). These traits make real-world temporal data irregular and hard to manage.

Several research communities address the challenge of irregular temporal data from different perspectives, as its analysis depends heavily on the task, application setting, and modeling approach. As a result, the problem spans multiple fields, including mobility analytics (da Silva et al., 2019), irregular time series (*ITS*) classification (Kidger et al., 2020), forecasting (Weerakody et al., 2021), and imputation (Luo et al., 2018; Li & Marlin, 2020), to name a few. Due to this vast amount of tasks, and despite some shared challenges, communities working on irregular temporal data tend to be separated, each relying on its own set of techniques, such as traditional statistical or data mining models (Hamilton, 2020), neural networks (Wang et al., 2024), or differential equations (Rubanova et al., 2019), often resulting in domain-specific tools and libraries. This is not inherently a drawback, but can lead to fragmented research efforts. The challenges of irregular temporal data are amplified in supervised learning, where standardized benchmarks are notably lacking. While repositories exist for *regular* time series classification (Dau et al., 2019), truly *irregular* datasets, capturing real-world missingness and variability, remain scarce. Researchers often resort to artificially manipulated datasets (Weerakody et al., 2021), introducing assumptions that overlook structural missingness tied to data collection (Mitra et al., 2023). As a result, and given that many studies rely on a narrow range of datasets, the generalizability of their methods often remains untested.

We bridge this gap by proposing `pyrregular` (`github.com/fspinna/pyrregular`), a unified framework for irregular time series. *(1)* We introduce a taxonomy of irregularities and a dataset structure in a common array format that improves interoperability across libraries while supporting the handling, visualization, and modeling of irregular time series using existing analysis methods. *(2)* We introduce the first standardized dataset repository for irregular time series classification, and *(3)* we leverage this repository to propose the first generalized benchmark for state-of-the-art classifiers from different research domains, in an effort to centralize research on this topic. Specifically, we curate 34 irregular time series datasets and evaluate 12 time series classifiers. Our goal is to empower users to seamlessly explore and evaluate a wide range of libraries to address the challenges of irregular temporal data.

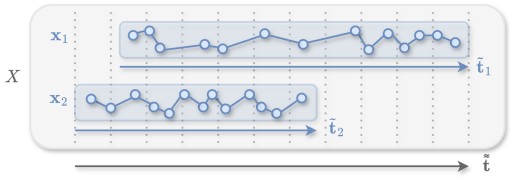

Figure 1: An example of an irregular time series, $\boldsymbol{X}$, comprising two signals $\mathbf{x}_1, \mathbf{x}_2$ with indices $\tilde{\mathbf{t}}_1, \tilde{\mathbf{t}}_2$, and the combined shared index $\tilde{\tilde{\mathbf{t}}}$.

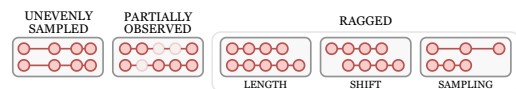

Figure 2: Different kinds of irregularity shown on a multivariate time series with 2 signals and containing up to 5 timestamps. Missing values are depicted as faded red if they were expected to be recorded, while they are omitted if they are caused by raggedness.

## 2 ORGANIZING IRREGULARITY

As our first contribution, we propose a systematic taxonomy that clearly distinguishes among different forms of irregularity. We begin by defining a time series signal.

**Definition 2.1** (Time Series Signal). A signal (or channel) is a sequence of $\tau$ observations, each associated to a timestamp, i.e., $\mathbf{x} = [(x_1, t_1), \ldots, (x_\tau, t_\tau)] = [x_{t_1}, \ldots, x_{t_\tau}] \in \mathbb{\mathring{R}}^\tau$.

A single signal can be *irregular* for two reasons: *uneven sampling*, when at least one interval $t_{k+1} - t_k$ differs from a constant $\Delta t$, and *partially observed*, when expected values are missing and marked as *NaN*. The set of real numbers extended with the *NaN* symbol is here represented as $\mathbb{\mathring{R}}$. We denote with $\tilde{\mathbf{t}} = [t_1, \ldots, t_\tau] \in \mathbb{R}^\tau$, the sorted collection of all timestamps where an observation of signal $\mathbf{x}$ was, or should have been recorded, and with $\tau = |\tilde{\mathbf{t}}|$ the number of observations.

**Definition 2.2** (Time Series). A time series is a collection of $d$ signals, $\boldsymbol{X} = \{\mathbf{x}_1, \ldots, \mathbf{x}_d\} \in \mathbb{\mathring{R}}^{d \times T}$.

Time series timestamps are the sorted union of all signal timestamps, i.e., $\tilde{\tilde{\mathbf{t}}} = \bigcup_{j=1}^d \tilde{\mathbf{t}}_j \in \mathbb{R}^T$, with $T = |\tilde{\tilde{\mathbf{t}}}|$, as shown in Figure 1. In addition to these intrinsic irregularities, tensor representations introduce a third, structural type: *raggedness*, that is the necessity of padding due to length, sampling, or alignment mismatches between signals. Hence, there are three independent irregularity causes: *uneven sampling*, *partial observation*, and *raggedness*, as depicted in Figure 2. While these categories have appeared informally in prior literature, here we show that they are independent: none implies the others. Unevenly sampled time series do not necessarily imply the presence of partially observed data, as seen in Figure 2 (left). This commonly happens in trajectory data, where the timestamps are usually highly uneven, but shared across the latitude and longitude signals. Vice versa, the presence of unobserved data does not imply uneven timestamps, as an observation may be accidentally missing from an overall constant sampling. Finally, neither unevenly sampled nor partially observed data imply raggedness. In particular, the two leftmost time series shown in Figure 2 could be stored in $2 \times 4$ and $2 \times 5$ matrices, respectively, without requiring any padding.

Raggedness arises because of different issues created when storing a multivariate time series in an array-like structure. As so, a single, univariate signal cannot be ragged by itself. In general, raggedness arises when at least two signals, $a$ and $b$, do not share the same timestamps, i.e., $\tilde{\mathbf{t}}_a \neq \tilde{\mathbf{t}}_b$. We identify three independent fundamental reasons why this can happen. The first is *ragged length*, when $a$ and $b$ have a different number of observations: $\tau_a \neq \tau_b$. The second is *shift*, where at least one signal starts and ends before another: $(t_{a,1} < t_{b,1}) \wedge (t_{a,\tau_a} < t_{b,\tau_b})$. The third is *ragged sampling*, when at least one element of the sampling intervals differs between two signals, i.e., $\Delta t_{a,k} \neq \Delta t_{b,k}$ for some $k$, where $\Delta t_{a,k} = t_{a,k+1} - t_{a,k}$ and $\Delta t_{b,k} = t_{b,k+1} - t_{b,k}$. Again, none of these, by itself, implies the other, as shown in Figure 2, and, in more detail, in Appendix B. Combinations of these issues yield highly irregular data, where *NaN* can indicate either a missing value in a partially observed time series or padding due to raggedness. Moreover, raggedness can exist also in a time series dataset, i.e., a collection of $n$ time series, $\mathbf{X} = \{\boldsymbol{X}_1, \ldots, \boldsymbol{X}_n\} \in \mathbb{\mathring{R}}^{n \times d \times \mathcal{T}}$, as all instances share the same sorted timestamps, $\mathbf{t} = \bigcup_{i=1}^n \tilde{\tilde{\mathbf{t}}}_i \in \mathbb{R}^{\mathcal{T}}$, with $\mathcal{T} = |\mathbf{t}|$. The *timestamp index* for the whole dataset is denoted as $\mathbf{k} = [1, \ldots, \mathcal{T}]$.

Associated with time series datasets are often *static attributes*, which refer to information linked to individual instances that remain independent of the time dimension. These attributes can also serve as targets in supervised tasks. Specifically, we focus on classification, i.e., targets are categorical.

## 3    RELATED WORK

**Datasets and Benchmarks.** There is a significant divide in the literature in the availability of datasets and benchmarking efforts, between *regular* and *irregular* time series data. Supervised learning for *regular* time series data is extensively addressed in the literature, with numerous "bake-offs" (Bagnall et al., 2017; Ruiz et al., 2021; Middlehurst et al., 2024b) benchmarking state-of-the-art classifiers on hundreds of standard datasets from the UEA and UCR repositories (Dau et al., 2019; Bagnall et al., 2018). On the contrary, the benchmarking literature on *irregular* time series remains limited. While secondary sources, such as (Weerakody et al., 2021; Wang et al., 2024), offer surveys on specific tasks like *ITS* imputation, comprehensive benchmarks for downstream tasks like classification are largely confined to primary studies (Kidger et al., 2020; Shukla & Marlin, 2021; Du et al., 2023). Even within these studies, evaluations are often performed on a small number of datasets. Moreover, benchmark datasets are not always inherently irregular; instead, they are commonly derived from regular datasets through simulation, i.e., dropping valid observations (Weerakody et al., 2021). Although this strategy can create *ITS*, introducing missingness is a non-trivial process requiring careful decisions about the type of missingness to simulate (Rubin, 1976). Adding to these challenges, a recent study (Mitra et al., 2023) highlighted that most research neglects structural missingness, referring to non-random, multivariate patterns of missingness within datasets. Such patterns can be faithfully preserved only by maintaining the original data with minimal modifications, which is the central focus of this proposal.

**Libraries.** Regarding *regular* time series data, Python libraries such as `sktime` (Löning et al., 2019), `aeon` (Middlehurst et al., 2024a), and `tslearn` (Tavenard et al., 2020) provide a wide range of classifier implementations, along with access to the UEA and UCR repositories, enabling systematic and reproducible evaluations. Although some of these datasets contain irregularities, the typical approach involves imputing missing values and discarding timestamps during downstream tasks. The most prominent Python library for *irregular* time series analysis is `pypots` (Du, 2023). `pypots` offers several classifiers, a few partially observed time series datasets, and provides an interface for adding missingness in regular datasets. A limitation of `pypots` is that it overlooks irregularity from uneven sampling, ignoring timestamps. It also operates within its own ecosystem, lacking interfaces for cross-library comparisons. This makes using *ITS* with libraries like `aeon` and `sktime` difficult, due to incompatible data formats and requirements, hindering standardization efforts. The primary reason for these challenges is the difficulty in managing *ITS* due to high dimensionality, missing values, and timestamps. Most libraries for time series prediction require dense 3D tensors to represent time series, signals, and identifiers (IDs), often demanding extensive padding and increased memory usage. To mitigate this, special arrays to represent missing values or variable-length instances are often used. For example, `numpy` masked arrays (Harris et al., 2020) indicate valid entries with masks but are memory-inefficient since they store both data and masks. Alternatives include `awkward` arrays (Pivarski et al., 2020), jagged `pytorch` arrays (Paszke et al., 2017), ragged `tensorflow` arrays (Abadi et al., 2015), `zarr`, `pyarrow`, or `sparse` arrays (Abbasi, 2018). Although efficient in managing varied-sized data, these structures cannot inherently handle timestamps. Forecasting libraries like `nixtla` or `gluonTS` (Alexandrov et al., 2020) typically use a *long format*, representing data as tuples $(i, j, t, x)$ with instance and signal IDs, timestamps, and observed values. While efficient for forecasting, this format requires pivoting for classification tasks, and static variables are either duplicated or stored separately, causing inefficiencies. Lastly, `xarray` (Hoyer & Hamman, 2017) supports timestamped multi-dimensional arrays but lacks native support for sparse *ITS*.

In summary, to the best of our knowledge, no existing array format is capable of representing irregular time series data in all their nuances. To address this limitation, we propose a framework that serves as a compatibility layer based on a unified array format, facilitating comprehensive benchmarking across a wide range of datasets and methods from diverse time series communities.

## 4    A UNIFIED FRAMEWORK FOR IRREGULAR TIME SERIES

This work addresses the gap in the literature on irregular time series by introducing an efficient container specifically designed for such data. This facilitates the integration of methods and datasets from various research communities into a unified framework. We outline key aspects of this solution. *(i) Ease of Use*: the framework supports several stages of the data science workflow, including visualization, preprocessing with classical and temporal slicing, and seamless conversion to dense arrays used in leading machine learning libraries. *(ii) Robustness*: the implementation leverages established

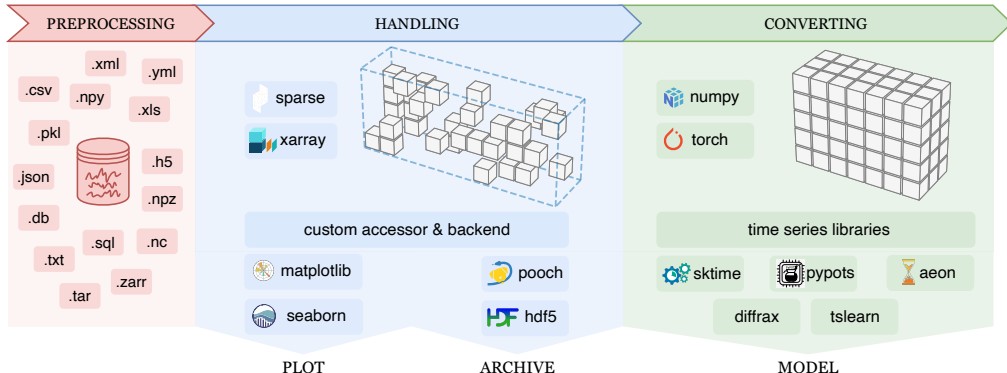

Figure 3: A simplified schema of our framework. (left) Data from different sources is preprocessed and represented in our proposed array container (center), which combines `xarray` with an underlying `sparse` tensor via a custom accessor and backend. This container can be easily manipulated, plotted, and stored. (right) Finally, it can also be converted into a more common dense representation, which can be used for downstream tasks with any standard time series library.

and well-maintained libraries, as there is no point in reinventing the wheel. *(iii) Flexibility*: the container supports several types of time series irregularities. *(iv) Replicability*: to ensure comparable results, preprocessing is standardized, addressing the variability in *ITS*. A depiction of the three steps of `pyrregular` is shown in Figure 3: *preprocessing*, where the original irregular data is transformed into our proposed container; *handling*, where the data can be explored, manipulated, and stored; and *converting*, where the data is prepared for downstream tasks. [1]

**Preprocessing.** The first step in our framework involves transforming raw irregular datasets into the proposed representation. *ITS* can be found in a wide variety of sources and formats (Figure 3, left), presenting unique challenges in terms of preprocessing. Regardless of the original data structure, our framework requires only a function capable of yielding the data in the standardized *long format*. In this representation, each row captures the time series ID, signal ID, timestamp, and observed value: $(i, j, t, x)$. The core intuition behind our approach is that the long format closely resembles the sparse coordinate (COO) representation (Duff et al., 2017).

The COO format, as implemented by `sparse` (Abbasi, 2018), can efficiently encode sparse 3D tensors, by using indices for the time series, signal, and timestamp, accompanied by an observed value entry, formally $(i, j, k, x)$. The key distinction between the long format and the COO representation lies in the handling of the timestamps: while the COO format requires discrete timestamp indices, $k$, the long format uses real-valued timestamps, $t$. An example is reported in Figure 4 (left). This difference, however, can be easily bridged by mapping the timestamps, $\mathbf{t}$, to discrete positions within the COO array, $\mathbf{k}$. Formally, given the timestamps vector $\mathbf{t} = [t_1, \ldots, t_{\mathcal{T}}]$, each timestamp can be mapped to its corresponding position (index), in the COO format as $\mathbf{k} = [1, \ldots, \mathcal{T}]$ (and vice-versa), as depicted in Figure 4 (center). With this mapping, converting between the long format and the COO representation can be easily accomplished, as the time series dataset is read once to construct the mapping and a second time to incrementally build the COO matrix by yielding each row as it is generated (Figure 4, right). Practitioners need only to define a custom function that, given their own data, incrementally produces rows in the long format. Even when the initial dataset is not organized in this manner, the conversion to the long format is typically straightforward. This process ensures uniformity across input formats and transparency, as the preprocessing steps are explicitly documented in this function, and can be reproduced at any time. Though it may be runtime-intensive, this step needs to be performed only once, after which the library streamlines all subsequent transformations and processing. The output after preprocessing is a sparse tensor, denoted as $\mathbf{X} \in \mathbb{\dot{R}}^{n \times d \times \mathcal{T}}$.

**Handling.** The COO representation offers advantages over the classical long format. First, it supports array-like operations with reasonable performance, including reshaping and slicing. Moreover, it allows for rapid conversion to task-specific array structures, such as other sparse formats like GCXS

---

[1]Code: `https://github.com/fspinna/pyrregular`. Examples are available in Appendix G.

(Shaikh & Hasan, 2015). Compared to classical dense arrays, its primary advantage lies in memory efficiency, as only the recorded observations are stored. All padding is represented by a *fill value* and remains implicit, meaning it is not directly stored but is generated only when the sparse array is transformed into a dense form. We propose setting such value to *NaN* to capture *raggedness*. Further, the COO format naturally accommodates partially observed data by explicitly storing a fill value. This allows for distinguishing between the two types of missing data previously discussed. Specifically, an explicitly stored fill value, i.e., a row $(i, j, k, NaN)$, can indicate a missing entry that should be present, while implicit *NaN*s reflect missingness due to data raggedness. In this sense, the COO tensor by itself is enough to represent both ragged and partially observed time series.

However, to capture an unevenly sampled time series, it is also essential to store the timestamps. To achieve this, we leverage the timestamp to COO ($\mathbf{t}$ to $\mathbf{k}$) mapping using xarray (Figure 3, center). In particular, we use xarray (Hoyer & Hamman, 2017) to store the timestamps and extend it to utilize an underlying sparse COO tensor. These functionalities are possible through our custom backend and accessor, which extend the xarray library, to support sparse arrays. Further, xarray naturally facilitates the storage of static attributes linked to any dataset dimension, such as class labels in classification tasks.

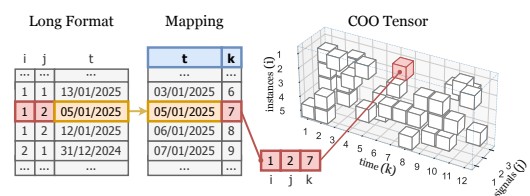

Figure 4: Long format to COO tensor conversion process. Each row of the long format is processed to retrieve the absolute position $k$ of a given timestamp $t$. The triplet, instance ID ($i = 1$), signal ID ($j = 2$), and timestamp index ($k = 7$), is used to populate the sparse COO tensor.

Overall, this approach offers significant storage efficiency, particularly given the typically high data sparsity (see Section 5.2), and ensures ease of use by supporting all existing xarray functions like timestamp range queries. Further, our accessor enables plotting, while our backend allows direct saving and loading to a hierarchical data format, locally or online, eliminating the need to perform the preprocessing step again.

**Converting.** Despite its advantages, xarray is not directly supported by most libraries for supervised learning tasks. Therefore, it is crucial to demonstrate how this array structure can be efficiently prepared for such applications[2]. Specifically, for classification tasks, $\mathbf{X} \in \dot{\mathbb{R}}^{n \times d \times \mathcal{T}}$ should be transformed into a dense tensor that minimizes raggedness while preserving the inherent missingness from partially observed time series and maintaining the order of observations within the same time series. This conversion is important because, in classification tasks, raggedness is typically irrelevant to the target and would otherwise result in vast dense arrays filled predominantly with *NaN*s. For instance, the specific starting dates of time series, such as $a$ beginning on January 23rd and $b$ on January 30th, are typically uninformative with respect to the output class, so we generally want to avoid introducing 7 leading *NaN*s in time series $b$ to account for the shift. For a COO array, this transformation corresponds to a dense ranking operation on the timestamp index, $k$, performed time series-wise. Formally, for each COO entry $(i, j, k, x)$, we produce $(i, j, rank_i(k), x)$, where:

$$rank_i(k) = 1 + |\{k' \in [1, T_i] : k' < k\}|.$$

This process shifts the timestamp indices within each time series, $\mathbf{X}_i$, into a consecutive sequence ranging from 1 to its length, $T_i$. As a result, the tensor $\mathbf{X} \in \dot{\mathbb{R}}^{n \times d \times \mathcal{T}}$ can be densified into a more compact, $\mathbf{X}' \in \dot{\mathbb{R}}^{n \times d \times T}$, where $T = \max_i^n (T_i)$. This ensures minimal raggedness, with the timestamp dimension set to the maximum number of timestamps in any time series. $\mathbf{X}'$ can be used by downstream libraries such as sktime (Löning et al., 2019), aeon (Middlehurst et al., 2024a), tslearn (Tavenard et al., 2020), pypots (Du, 2023) and diffrax (Kidger, 2021).

## 5 CLASSIFICATION BENCHMARKS

We present a comprehensive benchmark enabled by pyrregular, in which we evaluate 12 classifiers from a variety of time series libraries on a curated collection of 34 *ITS* datasets. We assess model performance from multiple perspectives, including dataset characteristics, robustness across irregularity types, and the potential for performance improvement through fine-tuning.

---

[2]We report a summary of the main formats used to represent regular and irregular time series in Appendix E

Table 1: Datasets used for our benchmarks, divided by irregularity type: unevenly sampled (US), partially observed (PO), unequal length (UL), shift (SH), ragged sampling (RS).

| | health | | | human activity recognition | | | | | | | | | | | | | mobility | | | | | | | | | sensor | | | other | | | | | synth |
|---|---|---|---|---|---|---|---|---|---|---|---|---|---|---|---|---|---|---|---|---|---|---|---|---|---|---|---|---|---|---|---|---|---|---|
| | MI3 | P12 | P19 | CT | GM1 | GM2 | GM3 | GP1 | GP2 | GX | GY | GZ | LPA | PAM | PGZ | SGZ | AN | AOC | APT | ARC | GS | MP | SE | TA | VE | DD | DG | DW | IW | JV | PGE | PL | SAD | ABF |
| **US** | ✓ | ✓ | ✓ | ✗ | ✗ | ✗ | ✗ | ✗ | ✗ | ✗ | ✗ | ✗ | ✗ | ✓ | ✓ | ✗ | ✓ | ✗ | ✗ | ✗ | ✓ | ✗ | ✓ | ✓ | ✓ | ✗ | ✗ | ✗ | ✗ | ✗ | ✓ | ✗ | ✗ | ✓ |
| **PO** | ✓ | ✓ | ✓ | ✗ | ✗ | ✗ | ✗ | ✗ | ✗ | ✗ | ✗ | ✗ | ✗ | ✗ | ✗ | ✗ | ✗ | ✗ | ✗ | ✗ | ✗ | ✗ | ✗ | ✗ | ✗ | ✓ | ✓ | ✓ | ✗ | ✗ | ✗ | ✗ | ✗ | ✗ |
| **UL** | ✓ | ✓ | ✓ | ✓ | ✓ | ✓ | ✓ | ✓ | ✓ | ✓ | ✓ | ✓ | ✓ | ✓ | ✓ | ✓ | ✓ | ✓ | ✓ | ✓ | ✓ | ✓ | ✓ | ✓ | ✓ | ✗ | ✗ | ✗ | ✓ | ✓ | ✓ | ✓ | ✓ | ✗ |
| **SH** | ✓ | ✓ | ✓ | ✗ | ✗ | ✗ | ✗ | ✗ | ✗ | ✗ | ✗ | ✗ | ✗ | ✓ | ✓ | ✗ | ✗ | ✗ | ✗ | ✗ | ✓ | ✗ | ✓ | ✗ | ✓ | ✗ | ✗ | ✗ | ✗ | ✗ | ✓ | ✗ | ✗ | ✗ |
| **RS** | ✓ | ✓ | ✓ | ✗ | ✗ | ✗ | ✗ | ✗ | ✗ | ✗ | ✗ | ✗ | ✗ | ✓ | ✓ | ✗ | ✓ | ✗ | ✗ | ✓ | ✓ | ✗ | ✓ | ✓ | ✓ | ✗ | ✗ | ✗ | ✗ | ✗ | ✓ | ✗ | ✗ | ✗ |

Table 2: Summary of evaluated classifiers.

| Library | | Model | Type | Domain |
|---|---|---|---|---|
| `aeon` | (Spinnato et al., 2024) | BORF | dictionary-based transform + LGBM classifier | regular, ragged |
| | | RIFC | interval-based transform + LGBM classifier | partially observed |
| `diffrax` | (Kidger et al., 2020) | NCDE | neural controlled differential equations | unevenly sampled |
| `pypots` | (Cao et al., 2018) | BRITS | bidirectional recurrent imputation network | partially observed |
| | (Che et al., 2018) | GRU-D | gated recurrent unit with decay | partially observed |
| | (Zhang et al., 2022) | RAINDROP | graph neural network | partially observed |
| | (Du et al., 2023) | SAITS | self-attention-based imputation transformer | partially observed |
| | (Wu et al., 2022) | TIMESNET | temporal 2d-variation inception | partially observed |
| `sktime` | (Ke et al., 2017) | LGBM | gradient boosted tree | tabular |
| | (Dempster et al., 2021) | ROCKET | kernel-based transform + LGBM classifier | regular |
| | (Bagheri et al., 2016) | SVM | support vector machine with distance kernel | regular, ragged |
| `tslearn` | (Sakoe & Chiba, 1978) | KNN | distance-based with dynamic time warping | regular, ragged |

**Datasets.** Following established repositories such as UEA and UCR, we compile a diverse collection of datasets that vary in size (small to large), length (short to long), and dimensionality (univariate to multivariate), ensuring broad representativeness. We solely focus on naturally irregular datasets, without artificially inducing irregularity (Tables 1 and 6). First, our collection contains widely used *ITS* classification datasets: *PhysioNet 2012* (P12) (Silva et al., 2012), *PhysioNet 2019* (P19) (Reyna et al., 2020), and the *MIMIC-III* (MI3) clinical database (Johnson et al., 2016) from the medical domain, as well as *Pamap2* (PAM) (Reiss & Stricker, 2012) for physical activity monitoring. Additionally, we include the 11 variable-length univariate time series classification problems (Guna et al., 2014; Caputo et al., 2018; Mezari & Maglogiannis, 2018; Gao et al., 2014) from (Bagnall et al., 2020), the 4 partially observed datasets (Ihler et al., 2006; City of Melbourne, 2020) from (Middlehurst et al., 2024b), and the 7 variable-length multivariate time series classification problems (de Souza, 2018; Williams et al., 2006; Chen et al., 2014; Kudo et al., 1999; Hammami & Bedda, 2010) from (Ruiz et al., 2021). We also provide datasets that, to the best of our knowledge, were never used in these kinds of benchmarks. These include data for trajectory classification of entities such as mammals (AN)(Ferrero et al., 2018), birds (SE) (Browning et al., 2018), and vehicles like buses and trucks (VE), taxis (Moreira-Matias et al., 2013) (TA) and combinations of the previous (Zheng et al., 2010) (GS). Further, we include a small dataset about the productivity prediction for garment employees (Imran et al., 2021) (PGE), and a human activity recognition dataset (Vidulin et al., 2010) (LPA). Finally, inspired by the classical Cylinder-Bell-Funnel benchmark (Saito, 1994) for regular time series classification, we introduce an irregular version called *Alembics-Bowls-Flasks* (ABF), in which the class depends on the skewness of the time sampling. Where available, we use the default train/test split for training and inference, else we set them based on each dataset description and original paper.

**Models.** The objective of these experiments is to benchmark methods capable of naturally handling *ITS* without introducing bias through imputation. For this reason, and to keep the benchmarks to a reasonable amount, we limit our evaluation to classifiers that inherently support irregular inputs and are available in the aforementioned libraries (Table 2 and Appendix C). As classical baselines, we use K-Nearest Neighbors (KNN) with Dynamic Time Warping (Sakoe & Chiba, 1978), a time series Support Vector Machine (SVM) with a Longest Common Subsequence (LCSS) kernel (Bagheri et al.,

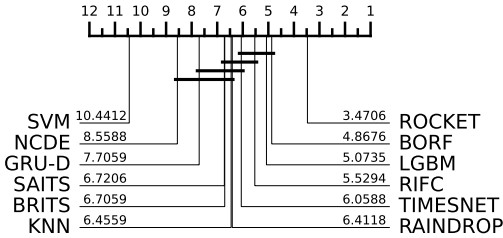
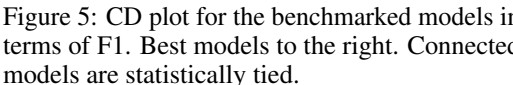
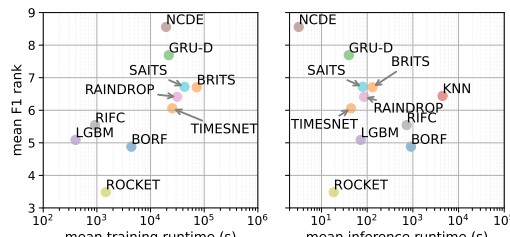

Figure 5: CD plot for the benchmarked models in terms of F1. Best models to the right. Connected models are statistically tied.

Figure 6: Mean F1 rank against training and inference runtimes for the top 11 models across all datasets. The best models are on the bottom left.

2016), and a LightGBM classifier (LGBM) trained directly on raw *ITS*, ignoring temporal dependencies. For regular time series models, we include the Bag-Of-Receptive-Fields (BORF) (Spinnato et al., 2024) from `aeon`, ROCKET (Dempster et al., 2020; 2021) via its MINIROCKET version in `sktime`, and a Random Interval Feature Classifier (RIFC). These models transform the data and rely on downstream classifiers; we use LGBM to handle possible *NaN*s. For partially observed data, we test GRU-D (Che et al., 2018), BRITS (Cao et al., 2018), RAINDROP (Zhang et al., 2022), a transformer, SAITS (Du et al., 2023), and an inception model, TIMESNET (Wu et al., 2023), from `pypots`, and a Neural Controlled Differential Equation model (NCDE) (Kidger et al., 2020) from `diffrax`.

**Experimental Setup.** Following standard practice in similar benchmarking studies (Bagnall et al., 2017; Middlehurst et al., 2024b), all models are trained using the default hyperparameters provided by their respective libraries or those recommended in the original papers. The goal of this benchmark, consistent with prior bake-offs, is to identify the model that best generalizes with a single, reasonable parameter configuration rather than fine-tuning each model for individual datasets. For this reason, the results of these benchmarks do not necessarily highlight the best possible model for a given task, but the model that generalizes best in many. Each model is allocated two weeks ($\approx 20000$ minutes) for training and inference on each dataset, with access to 32 cores and 512 GB of memory, and to a GPU when the model can use it[3]. Experiments are repeated three times for highly stochastic models, and the average performance is maintained. We use the F1 score with macro averaging as the primary performance metric, as it is robust in the presence of unbalanced data (Japkowicz, 2013), which occurs in some of our datasets. Accuracy results, along with additional metrics and statistical tests, are reported in Appendix D and are consistent with the following findings.

## 5.1 RESULTS AND DISCUSSION.

We present a comparative analysis of the aggregate results of the benchmark outcomes. We report a critical difference (CD) plot in Figure 5, which ranks models in terms of F1. Models are arranged from right to left, with lower ranks indicating better performance. Models connected by a horizontal bar are statistically tied under a one-sided Holm-corrected Wilcoxon signed-rank test with a significance threshold of 0.05. ROCKET emerged as the clear top-performing model, demonstrating consistent superiority across the datasets. Even if this result aligns with its established reputation as one of the best models for *regular* time series classification (Middlehurst et al., 2024b), its efficacy on *irregular* data is somewhat surprising, as ROCKET does not exploit any information about said irregularity. Following ROCKET, a cluster of methods, including BORF, LGBM, RIFC, TIMESNET, exhibits statistically tied performance. Lower ranks are occupied by RAINDROP, KNN, BRITS, followed by GRU-D and NCDE, with SVM distinctly identified as the worst-performing model.

**Performance vs. Time.** Besides predictive performance, runtime is also a significant factor. In Figure 6, we compare the average F1 rank against training and inference runtimes, discarding SVM for better readability. The better-performing, faster models appear in the bottom-left region of the plot. In terms of training, LGBM is the fastest, followed by RIFC and ROCKET, with ROCKET also being also very fast during inference. For this reason, ROCKET emerges as the best tradeoff between F1 and runtime. Interestingly, despite being designed for tabular data, LGBM performs

---

[3]System: IBM SYSTEM POWER AC922 Compute Nodes with $2 \times 16$-core 2.7GHz POWER9 CPUs, 512GB of RAM. NVIDIA Tesla V100 32GB GPU

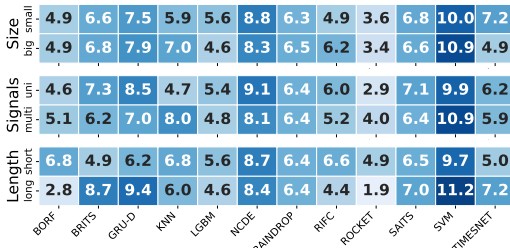
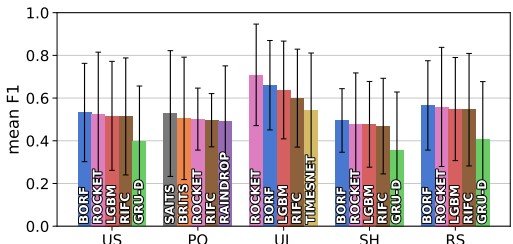

Figure 7: Mean F1 rank (lower is better) against dataset size in terms of instances (top), number of signals (center), and time series length (bottom).

Figure 8: Mean F1 (higher is better) of the 5 best-performing models for each type of irregularity.

well. This finding aligns with observations in (Tan et al., 2020), where gradient-boosting trees showed strong performance in *regular* time series *regression*. LGBM is a compelling choice due to its decent performance and exceptionally fast training time, making it attractive for practitioners needing solid baselines. Neural network-based methods, though designed for *ITS*, underperform in these bake-off-style benchmarks, except for their competitive inference runtime. Similar patterns appear in regular time series classification (Middlehurst et al., 2024b). We hypothesize that simpler, *generalist*, models, like ROCKET, excel in bake-off settings due to their low-variance, high-bias inductive bias, making them robust across a wide range of tasks, contrary to *specialized* models, which exhibit strong performance on specific types of irregularity or dataset characteristics, especially after fine-tuning.

**Performance vs. Dimension.** Figure 7 (top) shows the mean F1 ranks of all benchmarked models (lower is better), stratified by dataset size: small (at most 500 instances) and large (more than 500 instances). KNN and RIFC exhibit a noticeable worsening in rank on larger datasets, indicating limited scalability or reduced robustness as the number of training examples increases. In contrast, LGBM, and especially TIMESNET, improve significantly in rank, suggesting that more complex models, particularly inception-based ones, benefit from greater data availability to better exploit their capacity. Figure 7 (center) shows the mean F1 ranks for univariate and multivariate time series. While the best-ranked model is again ROCKET, all neural network-based approaches benefit from increased dimensionality, making them particularly suitable for multivariate time series. Figure 7 (bottom) reports the mean F1 ranks stratified by time series length: short (at most 360 observations) and long (more than 360 observations). Here, recurrent models such as GRU-D and BRITS, along with several other neural architectures, tend to struggle on longer sequences. RAINDROP stands out as an exception, likely owing to its graph-based design. Meanwhile, models that rely on localized or interval-based features, such as ROCKET, RIFC, and especially BORF, show improved performance on longer time series, indicating that in this case, simpler is better (more details available in Appendix C).

**Performance vs. Irregularity.** In Figure 8, we report the average F1 score of the top-5 performing models within each irregularity group (higher is better). ROCKET, BORF, and LGBM consistently rank among the top three across *unevenly sampled*, *unequal length*, *shifted*, and *ragged sampling* time series. GRU-D, while generally ranking lower overall, appears among the top five models in three out of the five groups, showing solid average performance. *Partially observed* time series exhibit markedly different behavior: here, models designed to handle missing data, such as SAITS and BRITS, outperform ROCKET, BORF, and LGBM. This suggests that explicitly modeling missingness can be highly beneficial, particularly for datasets with structured patterns of missing values.

**Performance after Fine-tuning.** In Table 3, we present the average performance of the top three *generalist* models, ROCKET, BORF, and LGBM, evaluated in terms of area under the Receiver Operating Characteristic curve (*auc*) and area under the Precision-Recall curve (*aupr*) following hyperparameter tuning. These evaluations follow the same 5-fold cross-validation setup and are compared against reference results from (Li et al., 2023; Liu et al., 2024; Zheng et al., 2024) on the two most commonly used irregular medical datasets: P12 (Silva et al., 2012) and P19 (Reyna et al., 2020). This benchmark aims to assess whether *generalist* classifiers can also be effectively fine-tuned for specific tasks, and to compare them with state-of-the-art *specialist* deep learning models such as CONTIFORMER (Chen et al., 2024), GRU-D (Che et al., 2018), MUSICNET (Liu et al., 2024), MTSFORMER (Zheng et al., 2024), and RAINDROP (Zhang et al., 2022). Results indicate that, when optimally fine-tuned, deep learning-based algorithms outperform simpler regular time series

Table 3: Comparison of best-performing models from the bake-off, against baseline reference results (higher is better). Best values in bold, second best underlined.

| | | BORF | CONTI FORMER | GRU-D | LGBM | MTS FORMER | MUSIC NET | RAIN DROP | ROCKET |
|---|---|---|---|---|---|---|---|---|---|
| **P12** | *auc* | 74.9±0.0 | 81.2±0.8 | 81.9±2.1 | 78.4±0.0 | 84.9±1.4 | **86.1±0.4** | 82.8±1.7 | 53.4±0.0 |
| | *aupr* | 33.4±0.0 | 43.9±3.0 | 46.1±4.7 | 38.1±0.0 | 51.1±3.7 | **54.1±2.2** | 44.0±3.0 | 15.8±0.0 |
| **P19** | *auc* | 80.1±0.0 | 79.2±2.3 | 83.9±1.7 | 85.2±0.0 | **88.8±1.5** | 86.8±1.4 | 87.0±2.3 | 77.3±0.0 |
| | *aupr* | 38.1±0.0 | 35.8±2.3 | 46.9±2.1 | 44.1±0.0 | **57.7±4.4** | 45.4±2.7 | 51.8±5.5 | 35.2±0.0 |

classifiers. However, except for ROCKET, which underperforms in this test, this advantage is not always substantial; for instance, LGBM achieves the fourth-best score on P19, outperforming models like CONTIFORMER and GRU-D. Another advantage of models such as ROCKET, BORF, and LGBM is that the performance is very stable, with near-zero standard deviation to a single decimal place. This underscores the value of being able to readily apply standard approaches, as they can offer fast, stable, and non-trivial baselines. However, deep learning offers more flexibility for optimizing on specific tasks, with reasonable inference times when aiming for raw performance for deployment purposes.

**Performance vs. Trustworthiness.** Though not the main focus of this work, we briefly address model trustworthiness, crucial in high-stakes fields like healthcare, where *ITS* are common. The most interpretable models in our benchmark are BORF, which relies on subsequence presence/absence, and RIFC, which uses simple interval-based features, both followed by a tree-based model. Neural models can be interpreted with gradient-based methods, though the reliability of their explanations on *ITS* is unexplored. The top-performing model, ROCKET, offers little interpretability and depends on expensive model-agnostic techniques (Theissler et al., 2022). Robustness to random initialization also matters: models with high variance across seeds hinder reproducibility. Stable methods like LGBM, BORF, and KNN may be preferable in sensitive settings, even at some cost in performance.

## 5.2 COMPLEXITY AND SCALABILITY

The end-to-end cost of the aforementioned classification pipelines in `pyrregular` consist of: *(i)* loading the dataset from disk into memory, *(ii)* converting it into a dense representation, and *(iii)* running the models. The latter component, previously discussed, is an *external* cost, since `pyrregular` wraps existing state-of-the-art classifiers from other libraries, and is reported separately in Table 9. In Table 4 we report instead the *internal* costs for datasets with a size greater than 10MB. The first two columns show the empirical times needed for dataset loading and conversion. Theoretically, the dominant cost arises when converting the sparse COO representation into dense form, which requires ranking the timestamps (Section 4). This amounts to sorting within each time series, leading to a complexity that scales linearly with the number of time series and log-linearly with the number of non-null observations per series. Thus, in practice, runtimes are efficient: for example, P19 takes around 3 seconds end-to-end, while the largest dataset, PA2, is converted in under one minute.

The third and fourth columns of Table 4 compare disk usage of our proposed array format with that of the raw data. In most cases, the proposed format either matches or substantially reduces disk requirements. For instance, GS decreases from 0.24GB in raw form to 0.09GB with our approach, while the reduction is even more pronounced for TA, which shrinks from 1.81GB to only 0.08GB. These reductions are especially valuable for large-scale datasets where disk I/O is a bottleneck.

The last three columns of Table 4 detail the memory footprint of different representations. The sparse COO representation (*ours*) incurs a cost of four times the number of non-null observations, accounting for the storage of coordinates and values. Conversion into a minimally ragged dense format leads to a worst-case memory complexity of $O(n \times d \times T)$, where $T = \max_i^n(T_i)$ is the longest series length. If the dataset is instead expanded into a fully ragged dense array, the worst-case complexity becomes $O(n \times d \times \mathcal{T})$, which grows quickly with irregularity. For example, on PA2, the sparse representation required only 3.93GB, compared to 5.33GB for a minimally ragged dense format. The largest savings are seen in highly irregular datasets: for TA, the sparse format used 0.34GB, while the fully ragged dense array would require over 4TB of memory, an impractical cost.

Table 4: Loading and conversion times (in seconds) for datasets using the proposed array format, along with disk size consumption (GB) compared to the raw data. Memory usage (GB) of the sparse representation is also reported relative to dense alternatives. Lower is better, best values in bold.

| | time (s) | | disk size (GB) | | memory (GB) | | |
| --- | --- | --- | --- | --- | --- | --- | --- |
| | loading | conversion | ours | raw | ours | dense min raggedness | dense with raggedness |
| ABF | 0.03 | 0.06 | ∼**0.00** | 0.01 | ∼**0.00** | ∼**0.00** | 0.81 |
| AOC | 0.09 | 0.12 | 0.01 | ∼**0.00** | 0.02 | **0.01** | **0.01** |
| APT | 0.30 | 0.42 | **0.02** | **0.02** | **0.08** | 0.11 | 0.11 |
| ARC | 0.21 | 0.28 | **0.01** | **0.01** | **0.05** | 0.14 | 0.14 |
| CT | 0.12 | 0.16 | **0.01** | **0.01** | 0.03 | **0.01** | **0.01** |
| GS | 1.25 | 3.62 | **0.09** | 0.24 | **0.29** | 8.58 | 377.15 |
| IW | 6.93 | 13.01 | 0.36 | **0.31** | 2.00 | **1.64** | **1.64** |
| LPA | 0.07 | 0.10 | ∼**0.00** | 0.02 | **0.01** | 0.07 | 4.02 |
| MI3 | 0.13 | 0.01 | ∼**0.00** | 0.04 | ∼**0.00** | ∼**0.00** | 0.03 |
| P12 | 0.35 | 0.65 | **0.01** | 0.08 | **0.1** | 0.45 | 6.35 |
| P19 | 0.96 | 2.08 | **0.03** | 0.24 | **0.31** | 3.41 | 3.43 |
| PA2 | 13.46 | 21.35 | **0.83** | 1.61 | **3.93** | 5.33 | 21.47 |
| PL | 0.06 | 0.07 | ∼**0.00** | ∼**0.00** | **0.01** | **0.01** | **0.01** |
| SAD | 0.49 | 0.65 | **0.02** | **0.02** | 0.14 | **0.08** | **0.08** |
| SE | 0.15 | 0.17 | **0.01** | 0.06 | 0.03 | **0.02** | 0.84 |
| TA | 1.49 | 2.34 | **0.08** | 1.81 | 0.34 | **0.22** | 4135.02 |
| VE | 0.05 | 0.08 | ∼**0.00** | 0.01 | **0.01** | **0.01** | 0.17 |

## 6 CONCLUSION

In this work, we presented `pyrregular`, a unified framework for addressing the challenges of *ITS*. By introducing a standardized repository for *ITS* classification and structuring the datasets in a common array format, we provided a cohesive way to work with varying forms of irregularity. Our extensive empirical evaluation of 12 state-of-the-art classifiers and baseline methods on 34 datasets emphasizes both the complexity of this domain and the benefits of a shared benchmarking resource. Results indicate that, with appropriate configuration and tuning, specialist models such as neural networks still attain state-of-the-art performance. However, extending their applicability across diverse tasks remains a significant challenge. Interestingly, simple generalist classifiers originally designed for regular time series data, such as ROCKET, perform remarkably well on irregular time series in bake-off-style benchmarks, even without leveraging the irregularity itself. This observation reveals a crucial research gap: the need to develop *generalist* methods capable of explicitly exploiting irregularities, such as missingness and timestamp information.

The construction of this extensive set of benchmarks was greatly facilitated by `pyrregular`, which abstracts the complexities of *ITS* across diverse libraries. While we aimed to provide a diverse and representative selection of baseline models, our choices were also guided by practical considerations such as library availability and interface compatibility, rather than exhaustive coverage. We acknowledge that several other relevant baselines could further enrich the comparison, such as emerging generalist temporal models, including LLM-based time series frameworks and multimodal pretraining approaches. Our goal was not to be fully comprehensive, but to establish a robust and extensible starting point for benchmarking within a unified framework. Further, we deliberately limited the scope of the benchmarks to classification, as achieving the same level of detail for other tasks, such as forecasting, anomaly detection, or imputation, would require an effort comparable in scale to what we present here, and is therefore left for future work. Nevertheless, because the proposed array format is task-independent and some curated datasets already include additional target variables, our framework naturally enables exploration of these tasks (see Appendix F for details). Going forward, `pyrregular` will be extended to such additional tasks and integrated with more datasets. In this direction, we will also explore dataset-specific analyses for complex clinical data, addressing missingness patterns and label imbalance. `pyrregular` will further be enriched with methods from a broader selection of time series libraries, increasing its relevance across diverse research domains.

ACKNOWLEDGMENTS

This study has been partially funded by the Italian Project Fondo Italiano per la Scienza FIS00001966 "MIMOSA", by the European Community Horizon 2020 programme under the funding schemes ERC-2018-ADG G.A. 834756 "XAI", G.A. 101070212 "FINDHR", G.A. 101120763 "TANGO", by the European Commission under the NextGeneration EU programme – National Recovery and Resilience Plan (Piano Nazionale di Ripresa e Resilienza, PNRR) Project: "SoBigData.it – Strengthening the Italian RI for Social Mining and Big Data Analytics" – Prot. IR0000013 – Av. n. 3264 del 28/12/2021, and M4C2 - Investimento 1.3, Partenariato Esteso PE00000013 - "FAIR" - Future Artificial Intelligence Research" - Spoke 1 "Human-centered AI".

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

## A  SUMMARY OF NOTATION

We have adopted a tensor-like notation inspired by (Kolda & Bader, 2009). The time series dataset is structured along three dimensions: the instance dimension, which consists of $n$ instances (e.g., $\boldsymbol{X}_i$ denotes the $i$-th time series in the dataset $\mathbf{X}$); the signal dimension, which includes $d$ channels (e.g., $\mathbf{x}_{i,j}$ represents the $j$-th signal in time series $\boldsymbol{X}_i$); and the time dimension, spanning $\mathcal{T}$ points (e.g., $x_{i,j,t_k}$ represents the $t_k$ observation of $j$-th signal in time series $\boldsymbol{X}_i$). We use tildes to specify the index being referenced (e.g., $t_k \in \mathbf{t}$ corresponds to the $k$-th timestamp at the dataset's level, while $t_k \in \tilde{\mathbf{t}}$ corresponds to the $k$-th timestamp at the time series's level). For improved readability, indices are omitted when they are not relevant.

Table 5: Summary of notation.

| Notation | |
|---|---|
| $\mathbf{X}, \boldsymbol{X}, \mathbf{x}, x$ | time series dataset, instance, signal, entry |
| $\mathbf{t}, \tilde{\mathbf{t}}, \tilde{\mathbf{t}}, t$ | timestamps for a time series dataset, instance, signal, entry |
| $\mathbf{k}$ | timestamp index |
| $n$ | number of instances in a dataset |
| $d$ | number of signals in a time series |
| $\mathcal{T}, T, \tau$ | number timestamps in a time series dataset, instance, signal |
| $i, j, k$ | indexes for instances, signals, timestamps |

## B  TAXONOMY OF TIME SERIES IRREGULARITIES

In addition to the well-known missingness taxonomy introduced in (Rubin, 1976) (MCAR, MAR, and MNAR), Mitra et al. (2023) proposed an additional category: structural missingness (SM). While Rubin's framework is typically formulated in terms of univariate patterns, SM highlights situations where missingness is organized across multiple variables and exhibits systematic structure. Our primary aim, distinct from previous works, is to preserve such structural patterns of missingness.

Consider, for instance, daily heart rate signals collected by wearables over three months. Data may be missing completely at random (MCAR) when some days are absent because the device randomly fails to sync, in which case missingness is unrelated to any variable. It may be missing at random (MAR) when data are more frequently absent on weekends, particularly for users with low recorded activity. It may be missing not at random (MNAR) when users remove the device precisely when feeling unwell, so missingness coincides with unrecorded spikes in heart rate. Finally, it may exhibit structural missingness (SM) when devices differ in recording frequency, such as once per second versus once per millisecond, or when a firmware update produces week-long gaps.

In this last case, missingness follows clear temporal patterns tied to device characteristics or design flaws, rather than to a single variable. Addressing such missingness (or raggedness) should therefore be an intentional modeling choice by the practitioner, not the result of routine preprocessing. We provide here formal definitions for each type of time series irregularity and use minimal counterexamples to show that none of these irregularities implies the others.

**Definition B.1** (Uneven Sampling). A signal $\mathbf{x} = [x_{t_1}, \ldots, x_{t_\tau}] \in \dot{\mathbb{R}}^\tau$ is said to be *unevenly sampled* if there exists at least one index $k \in \{1, \ldots, \tau - 1\}$ such that the time interval between successive observations is not constant, i.e., $t_{k+1} - t_k \neq \Delta t$ for some fixed $\Delta t \in \mathbb{R}$.

The same definition applies to time series instances and datasets, using their respective indices $\tilde{\mathbf{t}}, \mathbf{t}$.

**Definition B.2** (Partial Observation). A signal $\mathbf{x} = [x_{t_1}, \ldots, x_{t_\tau}] \in \dot{\mathbb{R}}^\tau$ is said to be *partially observed* if at least one value $x_{t_k}$ is missing and represented by a special symbol *NaN*, indicating the absence of an observation at a timestamp where one was expected, i.e., $x_{t_k} = \text{NaN}$ for some $k \in \{1, \ldots, \tau\}$.

Again, the same definition applies to time series instances and datasets.

**Definition B.3** (Raggedness). Raggedness is a structural irregularity that arises in a multivariate time series $\boldsymbol{X} = \{\mathbf{x}_1, \ldots, \mathbf{x}_d\} \in \dot{\mathbb{R}}^{d \times T}$ when the component signals do not share a common timestamp

index. Formally, raggedness is present when there exist at least two signals $\mathbf{x}_a$ and $\mathbf{x}_b$ such that $\tilde{\mathbf{t}}_a \neq \tilde{\mathbf{t}}_b$. It manifests in three independent forms:

- **(a) Ragged Length:** $\tau_a \neq \tau_b$.

- **(b) Shift:** $(t_{a,1} < t_{b,1}) \wedge (t_{a,\tau_a} < t_{b,\tau_b})$.

- **(c) Ragged Sampling:** $\Delta t_{a,k} \neq \Delta t_{b,k}$ for some $k$, where $\Delta t_{j,k} = t_{j,k+1} - t_{j,k}$. The index $k$ ranges from 1 to $\min(\tau_a, \tau_b) - 1$, so only intervals that exist in both signals are compared.

The same definition applies to time series datasets.

We now show that the five forms of time series irregularity are mutually independent: none implies any of the others. This is shown through minimal examples of time series that satisfy one irregularity condition while exhibiting none of the others.

## B.1 UNEVEN SAMPLING

Let $X = \{\mathbf{x}_a, \mathbf{x}_b\}$ be a time series where both signals share the same timestamp index, $\tilde{\tilde{\mathbf{t}}} = \tilde{\mathbf{t}}_a = \tilde{\mathbf{t}}_b = [t_1, t_2, t_3]$, and assume that the sampling intervals are not constant, i.e., $t_2 - t_1 \neq t_3 - t_2$. Then $X$ is unevenly sampled.

UNEVEN SAMPLING $\not\Rightarrow$ PARTIAL OBSERVATION. Suppose that all values in both $\mathbf{x}_a$ and $\mathbf{x}_b$ are observed (i.e., none are *NaN*). Then $X$ is unevenly sampled, but not partially observed.

UNEVEN SAMPLING $\not\Rightarrow$ RAGGEDNESS. Since $\tilde{\mathbf{t}}_a = \tilde{\mathbf{t}}_b$, both signals are aligned on the same timestamps. Therefore, $X$ is not ragged.

## B.2 PARTIAL OBSERVATION

Let $X = \{\mathbf{x}_a, \mathbf{x}_b\}$ be a time series where both signals share the same timestamp index, $\tilde{\tilde{\mathbf{t}}} = \tilde{\mathbf{t}}_a = \tilde{\mathbf{t}}_b = [t_1, t_2, t_3]$. Suppose that one observation is missing, e.g., $x_{a,t_2} = NaN$. Then $X$ is partially observed.

PARTIAL OBSERVATION $\not\Rightarrow$ UNEVEN SAMPLING. Let the timestamps be equally spaced, i.e., $t_2 - t_1 = t_3 - t_2 = \Delta t$. Then $X$ is partially observed but evenly sampled.

PARTIAL OBSERVATION $\not\Rightarrow$ RAGGEDNESS. Since both signals are defined over the same set of timestamps, $\tilde{\mathbf{t}}_a = \tilde{\mathbf{t}}_b$, $X$ is not ragged.

## B.3 RAGGED LENGTH

Let $X = \{\mathbf{x}_a, \mathbf{x}_b\}$ be a time series exhibiting ragged length, with $\tilde{\mathbf{t}}_a = [t_1, t_2]$ and $\tilde{\mathbf{t}}_b = [t_1, t_2, t_3]$. Then the unified timestamp index is $\tilde{\tilde{\mathbf{t}}} = [t_1, t_2, t_3]$, and $X$ satisfies $\tau_a = 2 \neq 3 = \tau_b$.

RAGGED LENGTH $\not\Rightarrow$ UNEVEN SAMPLING. Let the timestamps be evenly spaced, i.e., $t_2 - t_1 = t_3 - t_2 = \Delta t$. Then $X$ exhibits ragged length, but is evenly sampled.

RAGGED LENGTH $\not\Rightarrow$ PARTIAL OBSERVATION. Suppose that all values in both $\mathbf{x}_a$ and $\mathbf{x}_b$ are observed (i.e., no *NaN*s). Then $X$ exhibits ragged length, but is not partially observed.

RAGGED LENGTH $\not\Rightarrow$ SHIFT. Although the signals have different lengths, they both start at the same time, $t_1$. Hence, $X$ is not shifted.

RAGGED LENGTH $\not\Rightarrow$ RAGGED SAMPLING. The sampling intervals are identical across both signals, i.e., $\Delta \tilde{t}_{a,1} = \Delta \tilde{t}_{b,1} = t_2 - t_1$. Therefore, $X$ is not raggedly sampled.

## B.4 SHIFT

Let $X = \{\mathbf{x}_a, \mathbf{x}_b\}$ be a time series exhibiting shift, with $\tilde{\mathbf{t}}_a = [t_1, t_2]$ and $\tilde{\mathbf{t}}_b = [t_2, t_3]$. Then the unified timestamp index is $\tilde{\tilde{\mathbf{t}}} = [t_1, t_2, t_3]$, and $X$ is shifted, as $\mathbf{x}_a$ starts and ends before $\mathbf{x}_b$.

SHIFT $\not\Rightarrow$ UNEVEN SAMPLING. Let the timestamps be evenly spaced, i.e., $t_2 - t_1 = t_3 - t_2 = \Delta t$. Then $X$ exhibits shift, but is evenly sampled.

SHIFT $\not\Rightarrow$ PARTIAL OBSERVATION. Suppose that all values in both $\mathbf{x}_a$ and $\mathbf{x}_b$ are observed (i.e., no *NaN*s). Then $\boldsymbol{X}$ exhibits shift, but is not partially observed.

SHIFT $\not\Rightarrow$ RAGGED LENGTH. Both signals have the same number of observations, i.e., $\tau_a = \tau_b = 2$. Hence, $\boldsymbol{X}$ exhibits shift but not ragged length.

SHIFT $\not\Rightarrow$ RAGGED SAMPLING. The sampling intervals within each signal are equal, i.e., $\Delta \tilde{t}_{a,1} = t_2 - t_1 = \Delta \tilde{t}_{b,1} = t_3 - t_2$. Therefore, $\boldsymbol{X}$ is not raggedly sampled.

### B.5 RAGGED SAMPLING

Let $\boldsymbol{X} = \{\mathbf{x}_a, \mathbf{x}_b\}$ be a time series exhibiting ragged sampling, with $\tilde{\mathbf{t}}_a = [t_1, t_2]$ and $\tilde{\mathbf{t}}_b = [t_1, t_3]$. Then the unified timestamp index is $\tilde{\tilde{\mathbf{t}}} = [t_1, t_2, t_3]$, and the sampling intervals differ across signals: $\Delta \tilde{t}_{a,1} = t_2 - t_1 \neq t_3 - t_1 = \Delta \tilde{t}_{b,1}$.

RAGGED SAMPLING $\not\Rightarrow$ UNEVEN SAMPLING. Let the global timestamps satisfy $t_2 - t_1 = t_3 - t_2 = \Delta t$. Then $\boldsymbol{X}$ is raggedly sampled but not unevenly sampled.

RAGGED SAMPLING $\not\Rightarrow$ PARTIAL OBSERVATION. Suppose that all values in both $\mathbf{x}_a$ and $\mathbf{x}_b$ are observed (i.e., no *NaN*s). Then $\boldsymbol{X}$ exhibits ragged sampling, but is not partially observed.

RAGGED SAMPLING $\not\Rightarrow$ RAGGED LENGTH. Both signals contain the same number of observations, $\tau_a = \tau_b = 2$. Thus, $\boldsymbol{X}$ is not ragged in length.

RAGGED SAMPLING $\not\Rightarrow$ SHIFT. Both signals start at the same time, $t_1$, and have the same length. Therefore, $\boldsymbol{X}$ is not shifted.

These examples are minimal and can be easily extended to longer signals and time series. They suffice to establish that all forms of irregularity discussed, both in the main and raggedness subtypes, are pairwise independent. None of them implies any other, as illustrated also in Figure 2. To the best of our knowledge, this taxonomy accounts for all known forms of structural time series irregularity relevant to data modeling and representation.

## C  EXPERIMENTAL DETAILS.

In this section, we summarize experimental details regarding the models and datasets.

### C.1  MODELS

The objective of these experiments is to benchmark methods capable of naturally handling irregular time series without introducing bias through imputation techniques. To achieve this, we limit our evaluation to classifiers that inherently support missing data in their input and are available in major time series libraries. Below, we describe the implementation details and hyperparameters for each method. Parameters that are not mentioned are left to their default in their library implementation.

**Bag-of-Receptive-Fields** (BORF)  The Bag of Receptive Fields (BORF) algorithm (Spinnato et al., 2024) from the `aeon` library extracts discretized subsequences and counts their appearance in the time series, allowing the presence of missing data. A downstream LightGBM classifier with default parameters is used to handle transformed features. For the fine-tuned benchmark, the hyperparameter was on performed on the *min_window_to_signal_std_ratio* in the interval $[0, 0.2]$ with 0.05 increments.

**Bidirectional Recurrent Imputation for Time Series** (BRITS)  The BR*ITS* algorithm (Cao et al., 2018), also from the `pypots` library, employs a bidirectional recurrent network for imputing and classifying incomplete time series. It uses a hidden layer size of 256 and a batch size of 32. Training runs for up to 1000 epochs, with early stopping after 50 epochs of no improvement.

**Gated Recurrent Unit with Decay** (GRU-D)  The GRU-D model (Che et al., 2018), available in the `pypots` library, extends the Gated Recurrent Unit architecture by introducing decay mechanisms that account for missing data patterns. The recurrent hidden layer size is set to 256, with a batch size of 32. Training uses a maximum of 1000 epochs, with early stopping triggered after 50 epochs of no improvement.

**K-Nearest Neighbors with DTW** (KNN)  This baseline model employs the `tslearn` K-Nearest Neighbors algorithm, configured to use the Dynamic Time Warping (DTW) distance measure. DTW incorporates temporal alignment to handle time series of varying lengths effectively. The distance computation uses a Sakoe-Chiba band (Sakoe & Chiba, 1978) of 10 points, which limits the warping window to a fixed radius.

**LightGBM** (LGBM)  LightGBM (Ke et al., 2017) is a gradient-boosting framework optimized for speed and efficiency, and can naturally handle missing values. In this baseline, it is trained directly with default parameters on raw time series data transformed into a tabular format using the `sktime Tabularizer`. For the fine-tuned benchmark, hyperparameter optimization was conducted over a predefined search space that included the number of leaves *(num_leaves)* $\in \{31, 63, 127\}$, maximum tree depth *(max_depth)* $\in \{-1, 7, 10\}$, *(learning_rate)* $\in \{0.05, 0.1\}$, and the minimum number of samples per leaf *(min_data_in_leaf)* $\in \{20, 100\}$.

**Neural Controlled Differential Equation** (NCDE)  The Neural CDE model (Kidger et al., 2020), implemented via the `diffrax` library, learns continuous-time representations of time series data using differential equations. It employs an Euler solver with a maximum of 100 steps, with step size equal to the minimum time difference between any two adjacent observations, a hidden layer size of 8, and a width size of 32. Training uses a maximum of 1000 iterations, using Adam as optimizer, with a starting learning rate of 0.01, patience of 200 for early stopping, and a learning rate reduction factor of 0.5 after 50 stagnant iterations.

**Raindrop** (RAINDROP)  The Raindrop model (Zhang et al., 2022), a graph-based neural network from `pypots`, handles irregular time series by sending messages over graphs that are optimized for capturing time-varying dependencies among sensors. This model uses 2 layers, a feed-forward network size of 256, 2 attention heads, and a dropout rate of 0.3. Training employs a batch size of 32, with early stopping after 50 epochs of no improvement.

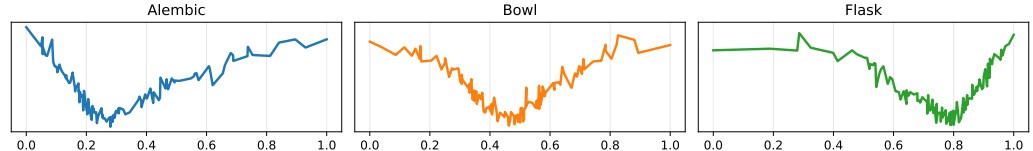

Figure 9: Three examples of instances from the (`ABF`) dataset, from left to right, Alembic, Bowl, and Flask.

**Random Interval Feature Classifier** (RIFC)  The Random Interval Feature Classifier (RIFC) leverages the `RandomIntervalFeatureExtractor` from the `sktime` library to generate simple statistical summaries (mean, standard deviation, minimum, maximum, median, skewness, and kurtosis) from randomly selected intervals within the time series, with the number of intervals being the logarithm of the time series length. These features are subsequently used by a downstream `LightGBM` classifier to perform classification.

**Minimally Random Convolutional Kernel Transform** (ROCKET)  Rocket, in its Minirocket implementation (Dempster et al., 2021) from the `sktime` library, employs 10000 fixed convolutional kernels to extract features from time series data. This implementation includes `MiniRocketMultivariateVariable`, which handles multivariate time series while tolerating missing data. The transformation could include missing data; therefore, instead of the most common ridge classifier, LightGBM with default parameters is used. For the fine-tuned benchmark, hyperparameter optimization was conducted over the number of kernels, *num_kernels* $\in$ $\{100, 500, 1000, 5000, 10000, 50000\}$.

**Self-Attention Imputation for Time Series** (SAITS)  The SA*ITS* model (Du et al., 2023), implemented in the `pypots` library, employs a transformer-based architecture specifically tailored for time series imputation. It utilizes a dual self-attention mechanism across temporal dimensions, enabling it to capture both global and local patterns despite missing values. In this configuration, SA*ITS* is trained with 2 attention layers, a model dimension of 256, 4 attention heads, and hidden dimensions $d_k = 64$, $d_v = 64$, and $d_{\text{ffn}} = 128$. A dropout rate of 0.1 is used for both the transformer blocks and attention layers. The model is optimized over a maximum of 1000 epochs, with early stopping triggered after 50 stagnant epochs. Training is performed with a batch size of 32.

**Support Vector Machine with LCSS Kernel** (SVM)  This method uses the `sktime` implementation of a Support Vector Machine, enhanced with the Longest Common Subsequence (LCSS) distance kernel (Bagheri et al., 2016). LCSS is robust to missing values and temporal distortions, as it matches time series subsequences with allowable gaps. The kernel uses a Sakoe-Chiba constraint with a radius of 10. Each time series is standardized using z-score normalization. The model is trained for a maximum of 1000 iterations.

**TimesNet** (TIMESNET)  TimesNet (Wu et al., 2023) is a modern inception-based architecture designed for multivariate time series modeling, emphasizing temporal receptive fields through learnable convolutional kernels. Its implementation here leverages 2 layers and 3 convolutional kernels with dynamic top-$k$ temporal selection. The model dimension is set to 64, with a feed-forward network size of 128. Training is conducted using a batch size of 32 over 1000 epochs, with early stopping after 50 epochs without validation improvement.

## C.2  DATASETS

The repository includes 34 datasets, each briefly described below, along with the data preparation steps applied. [4] For datasets without a predefined train-test split, we created a stratified, instance-based 70-30% train-test split.

---

[4]Data is hosted at `https://huggingface.co/datasets/splandi/pyrregular`.

**Alembics Bowls Flasks.** (`ABF`)  This dataset is inspired by the classical Cylinder-Bell-Funnel (CBF) benchmark (Saito, 1994) for regular time series classification. Similarly to CBF, there are three classes, which are Alembics, Bowls, and Flasks. The classes differ by how much the temporal axis is skewed, i.e., if it has positive (Alembic), negative (Flask), or no skewness (Bowl). For each time series, 128 values are sampled from a circumference and then standardized. There are 10 instances for each class in the training set and 300 for each in the test set. An example is presented in Figure 9.

**Animals** (`AN`)  This dataset, generated during the Starkey project (Ferrero et al., 2018), consists of trajectories from three animal species—elk, deer, and cattle. The classification task commonly used in the literature (Ferrero et al., 2018; Landi et al., 2023b;a) involves inferring the species based on movement patterns. The target classes in the dataset are balanced, with 38 trajectories for the elk, 30 for the deer, and 34 for the cattle.

**Geolife** (`GS`)  This dataset was collected during the GeoLife Project (Microsoft Research Asia) from April 2007 to August 2012 (Zheng et al., 2009; 2008; 2010). It contains the trajectories of 182 users and has been preprocessed as detailed in the public *User Guide-1.3*. One of the most common supervised machine-learning tasks using this dataset is to identify (a subset) of the 11 means of transportation. We defined three target variables with a decreasing number of classes. The first target variable includes all the means of transportation in the dataset: airplane, bike, boat, bus, car, motorcycle, run, subway, taxi, train, and walk. The second target variable, used in (Ferrero et al., 2018), groups the transportation modes into six classes: bike, bus/taxi, car, subway, train, and walk. The third target variable, used in (Landi et al., 2023b), simplifies the classification into two categories: private (bike, boat, car, motorcycle, run, walk) and public (the remaining modes of transportation). In Section 5, we benchmark the models against the first target variable. In this setting, each class accounts for approximately 9.1% of the total instances, but the standard deviation is 12.7%, i.e., the target variable is highly imbalanced.

**GPS Data of Seabirds** (`SE`)  This dataset, introduced in (Browning et al., 2018), consists of GPS data collected from 108 seabirds spanning three species: European shag (15), common guillemot (31), and razorbill (62). Similar to the *Animals* dataset, the species has been used to evaluate model performance in inferring species. The target variable is imbalanced, with the majority class (razorbill) comprising 62 individuals, while the minority class (European shag) includes only 15.

**Localization Data for Person Activity** (`LPA`)  Introduced in (Vidulin et al., 2010), this dataset contains data from 5 individuals performing 11 different actions: falling, lying, lying down, on all fours, sitting, sitting down, sitting on the ground, standing up from lying, standing up from sitting, standing up from sitting on the ground, walking. Each action was recorded by tracking the positions of the body's right and left ankles, chest, and belt in a 3-dimensional space, resulting in 12 distinct signals per time series.

**MIMIC-III Clinical Database Demo** (`MI3`)  Introduced by (Johnson et al., 2016; 2019) on the Physionet platform (Goldberger et al., 2000), the dataset contains health-related data associated with 40,000 patients in critical care at the Beth Israel Deaconess Medical Center from 2001 to 2012. Since the full version is available to credentialed users under strict requirements, we use the publicly available demo version in our work. We preprocess the data in accordance with (Harutyunyan et al., 2019). The classification target involves predicting in-hospital mortality.

**PAMAP2 Physical Activity Monitoring** (`PA2`)  This dataset, introduced in (Reiss & Stricker, 2012), contains data from 9 subjects (1 female, 8 male) performing 19 different physical activities: ascending stairs, car driving, computer work, cycling, descending stairs, folding laundry, house cleaning, ironing, lying, nordic walking, playing soccer, rope jumping, running, sitting, standing, transient, vacuum cleaning, walking, watching TV. The data includes measurements from 3 inertial measurement units (IMUs) positioned on the dominant arm, chest, and dominant side's ankle. Specifically, from each IMU sensor, the dataset contains information about the temperature, the 3-dimensional acceleration, gyroscope and magnetometer data, and the sensor orientation. Additionally, heart rate observations are included. The two types of sensors record data at different sampling rates: 100 Hz for the IMUs and 9 Hz for the heart rate monitor. We preprocess the data according to the authors' guidelines when downloading the dataset. Data from the "transient" activity, i.e., movements between the end of one

activity and the start of another, was excluded. The remaining 18 activities serve as classification target classes.

**PhysioNet 2012** (`P12`)  Published as data for the "Predicting Mortality of ICU Patients: The PhysioNet/Computing in Cardiology" challenge in 2012 (Silva et al., 2012), the data contains information about the patient, like age, gender, height, and weight, and 37 different types of time series. Similar to the MIMIC-III dataset, the classification target is about predicting in-hospital death.

**PhysioNet 2019** (`P19`)  Published as data for the "Early Prediction of Sepsis from Clinical Data: The PhysioNet/Computing in Cardiology" challenge in 2019 (Reyna et al., 2020), the dataset contains demographic information about the patients, such as age, gender, height, and weight, alongside 34 other time-series variables for vital signs and laboratory test values. The classification task involves predicting whether a patient has sepsis or not.

**Productivity Prediction of Garment Employees** (`PGE`)  Introduced in (Imran et al., 2021), this dataset contains information about garment manufacturing processing on a per-team level. Additionally, this dataset contains a team productivity performance index, which ranges between 0 and 1. As suggested by the authors, we use this index as a classification target. Specifically, we defined a team *efficient* if the productivity performance index is strictly greater than $0.75$.

**Taxi** (`TA`)  This dataset, introduced as part of the "ECML/PKDD 15: Taxi Trip Time Prediction (II) Competition" (Moreira-Matias et al., 2013) consists of 121,312 trajectories of Taxis in Porto (Portugal). The classification task is to predict the type of call that generated the run. The types of calls could be: *A* if this trip was dispatched from the central, *B* if this trip was demanded directly to a taxi driver on a specific stand *C* otherwise. The classes are balanced.

**Vehicles** (`VE`)  GPS trajectories about two different types of vehicles -buses and trucks- moving in Athens. This dataset is available from download from the Chorochronos Archive (ChoroChronos Archive).

**UEA and UCR Irregular Datasets.**  The other 22 irregular time-series datasets were downloaded from the UEA and UCR dataset repository. In particular, we included the following datasets:

- 11 variable-length univariate time series classification problems from (Bagnall et al., 2020): AllGestureWiimoteX, AllGestureWiimoteY and AllGestureWiimoteZ (`GX`, `GY`, `GZ`) from (Guna et al., 2014); GestureMidAirD1, GestureMidAirD2, and GestureMidAirD3 (`GM1`, `GM2`, `GM3`) from (Caputo et al., 2018); GesturePebbleZ1 and GesturePebbleZ2 (`GP1`, `GP2`) from (Mezari & Maglogiannis, 2018); PickupGestureWiimoteZ and ShakeGestureWiimoteZ (`PGZ`, `SGZ`) from (Guna et al., 2014); PLAID (`PL`) from (Gao et al., 2014);

- 4 fixed length univariate time series with missing values from (Middlehurst et al., 2024b): DodgerLoopDay, DodgerLoopGame, and DodgerLoopWeekend (`DD`, `DG`, `DW`) from (Ihler et al., 2006); MelbournePedestrian (`MP`) (City of Melbourne, 2020) extracted from the City of Melbourne website;

- 7 variable-length multivariate time series from (Ruiz et al., 2021): AsphaltObstaclesCoordinates, AsphaltPavementTypeCoordinates, and AsphaltRegularityCoordinates (`AOC`, `APT`, `ARC`) from (de Souza, 2018); CharacterTrajectories (`CT`) from (Williams et al., 2006); InsectWingbeat (`IW`) from (Chen et al., 2014); JapaneseVowels (`JV`) from (Kudo et al., 1999); SpokenArabicDigits (`SAD`) from (Hammami & Bedda, 2010);

Table 6 contains the full list of curated datasets at the moment of publication on our repository. The list additionally contains some information about the datasets: the number of instances, #Inst, number of signals, #Sign, and number of observations, #Obs ($\max_i^n(T_i)$), the number of target classes #TC and the standard deviation between the number of instances per class (CU). Additionally, the dataset contains information about the time series, like the percentage of missing values (MV)-computed as the ratio between the *NaN* observations divided by the total number of observations- and the sampling coefficient of variation (SCV), alongside information on the different kind of irregularity in the dataset.

Table 6: Summary of dataset characteristics: the number of instances (#Inst), signals (#Sign), and observations (#Obs); target classes (#TC) and class imbalance (CU); as well as time-series-specific metrics like missing values (MV) and sampling coefficient of variation (SCV), and each type of irregularity, i.e., unevenly sampled (US), partially observed (PO), unequal length (UL), shift (SH), ragged sampling (RS).

| Cat | Name | Source | #Inst | #Sign | #Obs | #TC | CU (σ) | MV (%) | SVC | US | PO | UL | SH | RS |
|---|---|---|---|---|---|---|---|---|---|---|---|---|---|---|
| *health* | MI3 | (Johnson et al., 2016) | 57 | 17 | 145 | 2 | 0.20 | 0.83 | 0.60 | ✓ | ✓ | ✓ | ✓ | ✓ |
| | P12 | (Silva et al., 2012) | 7990 | 37 | 203 | 2 | 0.36 | 0.94 | 0.59 | ✓ | ✓ | ✓ | ✓ | ✓ |
| | P19 | (Reyna et al., 2020) | 40334 | 34 | 334 | 2 | 0.43 | 0.98 | 0.18 | ✓ | ✓ | ✓ | ✓ | ✓ |
| *human activity recognition* | CT | (Williams et al., 2006) | 2858 | 3 | 182 | 20 | 0.01 | 0.34 | 0.00 | ✗ | ✗ | ✓ | ✗ | ✗ |
| | GM1 | (Caputo et al., 2018) | 338 | 1 | 360 | 26 | 0.00 | 0.54 | 0.00 | ✗ | ✗ | ✓ | ✗ | ✗ |
| | GM2 | (Caputo et al., 2018) | 338 | 1 | 360 | 26 | 0.00 | 0.54 | 0.00 | ✗ | ✗ | ✓ | ✗ | ✗ |
| | GM3 | (Caputo et al., 2018) | 338 | 1 | 360 | 26 | 0.00 | 0.54 | 0.00 | ✗ | ✗ | ✓ | ✗ | ✗ |
| | GP1 | (Mezari & Maglogiannis, 2018) | 304 | 1 | 455 | 6 | 0.01 | 0.52 | 0.00 | ✗ | ✗ | ✓ | ✗ | ✗ |
| | GP2 | (Mezari & Maglogiannis, 2018) | 304 | 1 | 455 | 6 | 0.01 | 0.52 | 0.00 | ✗ | ✗ | ✓ | ✗ | ✗ |
| | GX | (Guna et al., 2014) | 1000 | 1 | 385 | 10 | 0.00 | 0.68 | 0.00 | ✗ | ✗ | ✓ | ✗ | ✗ |
| | GY | (Guna et al., 2014) | 1000 | 1 | 385 | 10 | 0.00 | 0.68 | 0.00 | ✗ | ✗ | ✓ | ✗ | ✗ |
| | GZ | (Guna et al., 2014) | 1000 | 1 | 385 | 10 | 0.00 | 0.68 | 0.00 | ✗ | ✗ | ✓ | ✗ | ✗ |
| | LPA | (Vidulin et al., 2010) | 273 | 12 | 2870 | 11 | 0.00 | 0.95 | 9.04 | ✓ | ✗ | ✓ | ✓ | ✓ |
| | PAM | (Reiss & Stricker, 2012) | 124 | 52 | 110883 | 16 | 0.03 | 0.82 | 0.01 | ✓ | ✓ | ✓ | ✓ | ✓ |
| | PGZ | (Guna et al., 2014) | 100 | 1 | 361 | 10 | 0.00 | 0.60 | 0.00 | ✗ | ✗ | ✓ | ✗ | ✗ |
| | SGZ | (Guna et al., 2014) | 100 | 1 | 385 | 10 | 0.00 | 0.57 | 0.00 | ✗ | ✗ | ✓ | ✗ | ✗ |
| *mobility* | AN | (Ferrero et al., 2018) | 102 | 2 | 291 | 3 | 0.03 | 0.50 | 1.21 | ✓ | ✗ | ✓ | ✗ | ✓ |
| | AOC | (de Souza, 2018) | 781 | 3 | 736 | 4 | 0.03 | 0.59 | 0.00 | ✗ | ✗ | ✓ | ✗ | ✗ |
| | APT | (de Souza, 2018) | 2111 | 3 | 2371 | 3 | 0.06 | 0.83 | 0.00 | ✗ | ✗ | ✓ | ✗ | ✗ |
| | ARC | (de Souza, 2018) | 1502 | 3 | 4201 | 2 | 0.01 | 0.91 | 0.00 | ✗ | ✗ | ✓ | ✗ | ✗ |
| | GS | (Zheng et al., 2010) | 5977 | 2 | 96282 | 11 | 0.13 | 0.99 | 10.27 | ✓ | ✗ | ✓ | ✓ | ✓ |
| | MP | (City of Melbourne, 2020) | 3633 | 1 | 24 | 10 | 0.00 | 0.00 | 0.01 | ✗ | ✗ | ✓ | ✗ | ✗ |
| | SE | (Browning et al., 2018) | 108 | 4 | 6048 | 3 | 0.18 | 0.60 | 0.00 | ✓ | ✗ | ✓ | ✓ | ✓ |
| | TA | (Moreira-Matias et al., 2013) | 121312 | 2 | 119 | 3 | 0.13 | 0.61 | 0.00 | ✓ | ✗ | ✓ | ✓ | ✓ |
| | VE | (ChoroChronos Archive) | 381 | 2 | 1095 | 2 | 0.22 | 0.57 | 5.29 | ✓ | ✗ | ✓ | ✗ | ✓ |
| *sensor* | DD | (Ihler et al., 2006) | 158 | 1 | 288 | 7 | 0.01 | 0.01 | 0.00 | ✗ | ✓ | ✗ | ✗ | ✗ |
| | DG | (Ihler et al., 2006) | 158 | 1 | 288 | 2 | 0.02 | 0.01 | 0.00 | ✗ | ✓ | ✗ | ✗ | ✗ |
| | DW | (Ihler et al., 2006) | 158 | 1 | 288 | 2 | 0.21 | 0.01 | 0.00 | ✗ | ✓ | ✗ | ✗ | ✗ |
| *other* | IW | (Chen et al., 2014) | 50000 | 200 | 22 | 10 | 0.00 | 0.70 | 0.00 | ✗ | ✗ | ✓ | ✗ | ✗ |
| | JV | (Kudo et al., 1999) | 640 | 12 | 29 | 9 | 0.03 | 0.46 | 0.00 | ✗ | ✗ | ✓ | ✗ | ✗ |
| | PGE | (Imran et al., 2021) | 24 | 9 | 59 | 2 | 0.13 | 0.19 | 0.68 | ✓ | ✗ | ✓ | ✓ | ✓ |
| | PL | (Gao et al., 2014) | 1074 | 1 | 1344 | 11 | 0.05 | 0.76 | 0.00 | ✗ | ✗ | ✓ | ✗ | ✗ |
| | SAD | (Hammami & Bedda, 2010) | 8798 | 13 | 93 | 10 | 0.00 | 0.57 | 0.00 | ✗ | ✗ | ✓ | ✗ | ✗ |
| *synth* | ABF | *new!* | 930 | 1 | 128 | 3 | 0.00 | 0.00 | 1.95 | ✓ | ✗ | ✗ | ✗ | ✗ |

Given $\mathbf{y}_h$ as the labels vector containing only the $h$-th class, CU is defined as follows:

$$CU = \sqrt{\frac{\sum_{h=0}^{c}(y_h - \mu)}{c}} \tag{1}$$

where $\mu$ is the average number of observations. Given $\Delta\tilde{\mathbf{t}}$ as the vector of differences between consecutive timestamps of a signal, the SCV is computed as the coefficient of variation (the ratio of the standard deviation to the mean) for each signal, averaged first across each time series and then over the entire dataset.

We divided the dataset into 6 categories based on the type of phenomena captured: *healthcare*, *human activity recognition*, *mobility* (or more generically, geo-temporal motion), *sensors*, *synthetic* data, and *others* for datasets that don't fall in any of the previous categories (like the UCR audio and speech categories).

# D  ADDITIONAL RESULTS AND STATISTICAL TESTS

The full result table in terms of F1 is available in Table 8. Further, we provide several other statistical tests, using a diverse range of metrics, and with respect to different dataset subgroups.

**Critical Difference Plots.**    Figure 10 shows the CD-plots for common performance metrics and runtimes. *F1*, *accuracy*, *roc-auc*, *precision*, and *recall* yield consistent rankings for the top four models, ROCKET, BORF, LGBM, and RIFC, as well as for the three lowest-performing ones: GRU-D, NCDE, and SVM. In the mid-range, rankings vary slightly across metrics: for instance, KNN performs notably worse in terms of F1 compared to accuracy, whereas TIMESNET shows the opposite trend. As for training time, KNN, being a lazy learner, is the fastest, followed by RIFC and ROCKET. Although LGBM ranks fourth, the previous results in median runtime (Figure 6) suggest that it may be slightly slower on smaller datasets but highly efficient on larger ones, which contributes to its overall strong performance. Neural network-based models generally exhibit longer training times but benefit from faster inference; nevertheless, ROCKET and LGBM maintain a performance edge across both phases.

*F1* CD-plots computed for subsets of datasets with specific characteristics, are shown in Figure 11. These plots provide additional and complementary insights to those in Figures 7 and 8. Notably, they reinforce the observation that models explicitly designed for partially observed data tend to outperform more general-purpose approaches, even though the top rankings remain closely contested among SAITS, RIFC, LGBM, BRITS, and ROCKET. BRITS and TIMESNET, in particular, show strong performance on shorter datasets, ranking second and third, respectively, and closely trailing ROCKET. The remaining plots are similar to those discussed in Section 5.

**Multiple Comparison Matrices.**    While the widely used CD-plot is effective, it has been criticized in (Ismail-Fawaz et al., 2023) for its susceptibility to manipulation, as the average rank of a model can be influenced by the performance of other comparators. For this reason, we also propose MCM matrix for several metrics in Figures 12 to 14. However, in our case, results are consistent with the CD-plots presented in the previous paragraph, and in the main text, and are presented here in the appendix only due to space limitations. Again, the top four models are always ROCKET, BORF, LGBM, and RIFC, and the lowest-performing are GRU-D, NCDE, and SVM, with mid-range models rankings changing slightly from metric to metric.

**Additional Performance vs Runtime Plots.**    We report in Figures 15 to 19 the performance rankings across multiple metrics, dataset subsets, and with respect to both training and inference times. In addition to the insights discussed in the main text, these figures reveal that neural network-based models tend to cluster together in terms of both runtime and performance, regardless of the dataset subset or evaluation metric. This suggests that, although their relative rankings may vary, their overall behavior remains consistent.

**Rank Correlation.**    We report in Figure 20 the F1 rank correlation among models. Models are hierarchically clustered using average linkage applied to the rank correlation matrix. Positive correlations indicate that models tend to perform similarly across datasets, reflecting comparable strengths or weaknesses, while negative correlations suggest divergent performance, highlighting complementary behaviors or differing inductive biases. Reinforcing the categorization proposed in the main text, the plot reveals a strong cluster of generalist methods, LGBM, ROCKET, RIFC, and BORF, which group together at the top hierarchical level. The second major cluster includes the remaining models, with specialist approaches like BRITS and GRU-D showing high correlation, which is expected given their shared RNN architecture. Similarly, TIMESNET and SAITS also form a coherent transformer/inception-style subgroup. Notable exceptions to the generalist/specialist categorization are SVM, likely due to its overall poor performance across datasets, and KNN, which we hypothesize behaves differently due to its lazy learning paradigm based on distances, which could be more prone to sensitivity to dataset-specific characteristics.

**Model Failures and Limitations.**    From these experiments, several model weaknesses become apparent, particularly in relation to specific data characteristics. For example, Figure 7 highlights how RNN-based methods fail to handle long time series effectively, while Table 3 shows that ROCKET underperformed relative to its baseline results after fine-tuning.

Additional insights arise from the CD plots in Figure 11. Comparing the general rankings in Figure 11a with those on specific subsets reveals which models are most sensitive to dataset properties. For instance, Figure 11g shows that the inception-based TIMESNET performs worse on smaller datasets, a point also observerd in Section 5. BORF, despite its strong overall performance, ranks third-to-last on partially observed data and declines significantly on short time series (Figures 11c and 11i). KNN also struggles under shift and ragged sampling conditions (Figures 11e and 11f). Notably, KNN was the weakest model in terms of memory consumption, which explodes with longer series (Table 9).

To provide a more fine-grained view, we report in Table 7 each model's worst performance in terms ratio between that worst-case rank and its average rank across all datasets. Higher ratios indicate greater variability, a phenomenon most pronounced among models that otherwise perform strongly on average, such as ROCKET, BORF, and LGBM. Several notable cases emerge. ROCKET, for instance, performs poorly on ABF, a dataset with highly uneven sampling. Similarly, BORF ranks 2.4 times worse than its average on the Mimic3 dataset, which is also highly irregular. Interestingly, LGBM performs unexpectedly poorly on the Garment dataset, whose small size would normally favor tree-based models.

These findings highlight that strong average performance does not necessarily imply robustness across all dataset types. In particular, models often fail on datasets with structural irregularities or atypical sampling patterns.

Table 7: Worst-case dataset performance for each model, along with the ratio between its worst rank and average rank across all datasets. Higher ratios indicate greater variability compared to average performance.

| model | worst dataset performance | worst-to-average rank ratio |
|---|---|---|
| BORF | Mimic3 | 2.4 |
| BRITS | AllGestureWiimoteX | 1.8 |
| GRU-D | CharacterTrajectories | 1.6 |
| KNN | Physionet2012 | 1.9 |
| LGBM | Garment | 2.3 |
| NCDE | ShakeGestureWiimoteZ | 1.4 |
| RAINDROP | InsectWingbeat | 1.8 |
| RIFC | GeolifeSupervised | 2.2 |
| ROCKET | Abf | 3.0 |
| SAITS | Animals | 1.5 |
| SVM | AllGestureWiimoteY | 1.2 |
| TIMESNET | DodgerLoopDay | 2.0 |

**Impact of irregularity on explanations.** As discussed in Section 5, XAI for irregular time series remains largely unexplored. `pyrregular` allows researchers to work directly with data while preserving its irregularities, avoiding the bias introduced by imputation choices, which is fundamental since explanations are known to be highly sensitive to input variations (Yeh et al., 2019). This, however, is only a first step. Even when the data retains its irregularity (as in our approach), and even when models can handle irregular inputs, the explainers themselves typically cannot. In line with the observations of Cinquini et al. (2023), we argue that this is primarily an implementation gap on the explainer side. Addressing this limitation would enable our taxonomy of irregularities to be applied to more fine-grained interpretability. For example, it could help distinguish whether a model assigns importance to a missing value because of partial observation or because of raggedness, offering deeper insights into the model's behavior under irregular conditions.

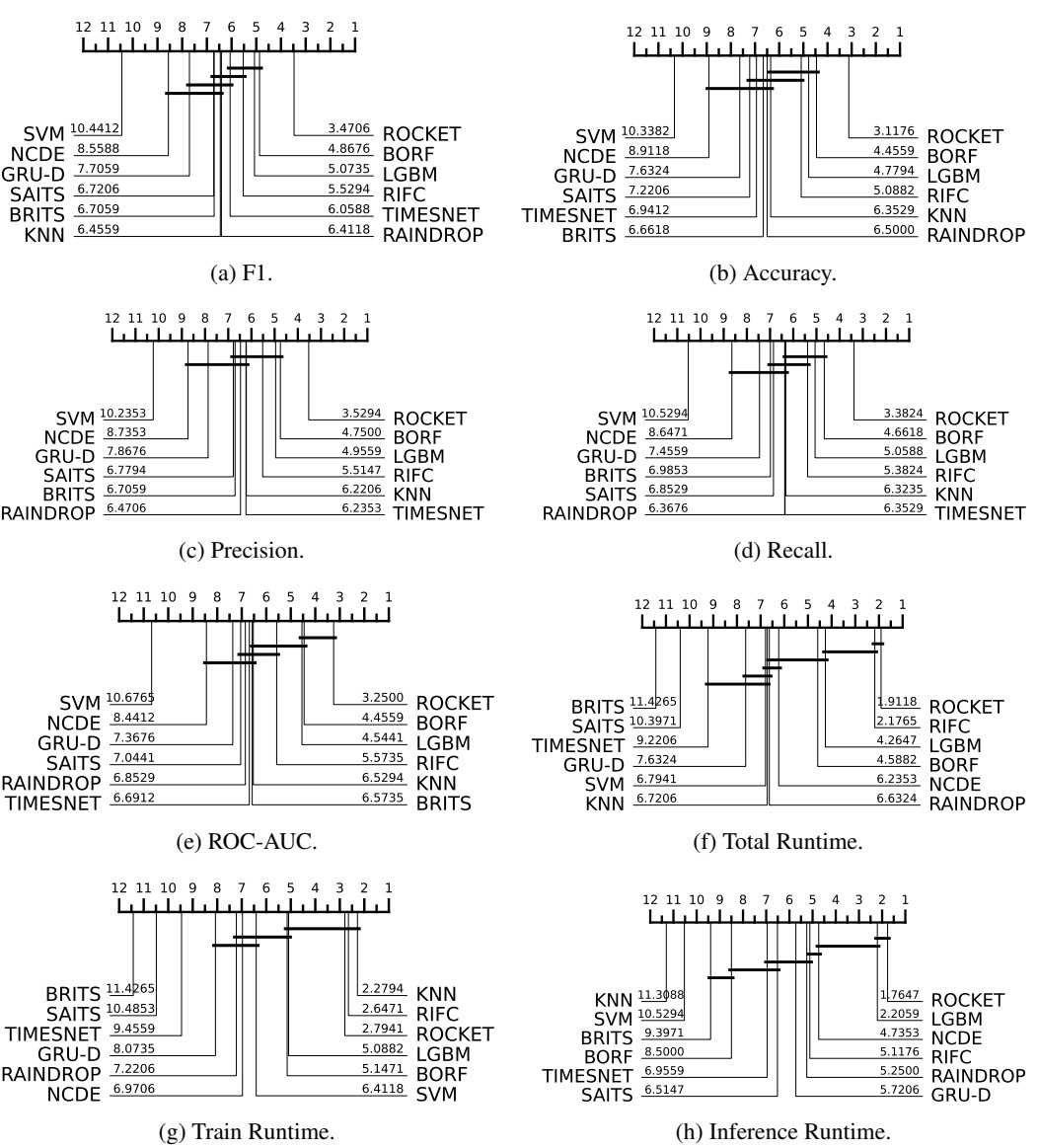

Figure 10: Critical Difference plot for the benchmarked models in terms of different metrics, for all datasets. Best models to the right. The performance of models connected by the bar is statistically tied, using a one-sided Holm-corrected Wilcoxon sign rank test with a critical value of $0.05$.

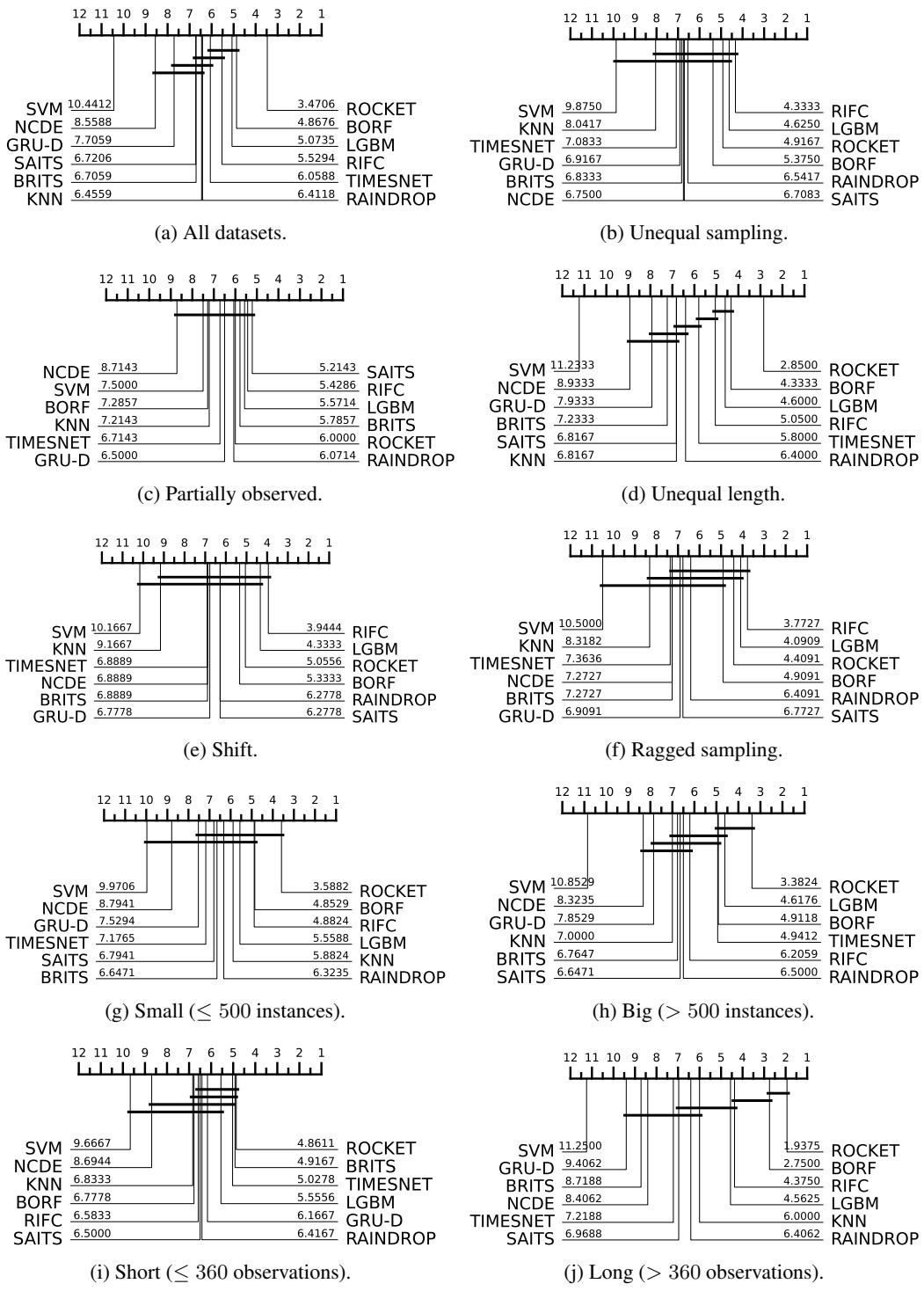

Figure 11: Critical Difference plot for the benchmarked models in terms of F1, divided into different groups. Best models to the right. The performance of models connected by the bar is statistically tied, using a one-sided Holm-corrected Wilcoxon sign rank test with a critical value of 0.05.

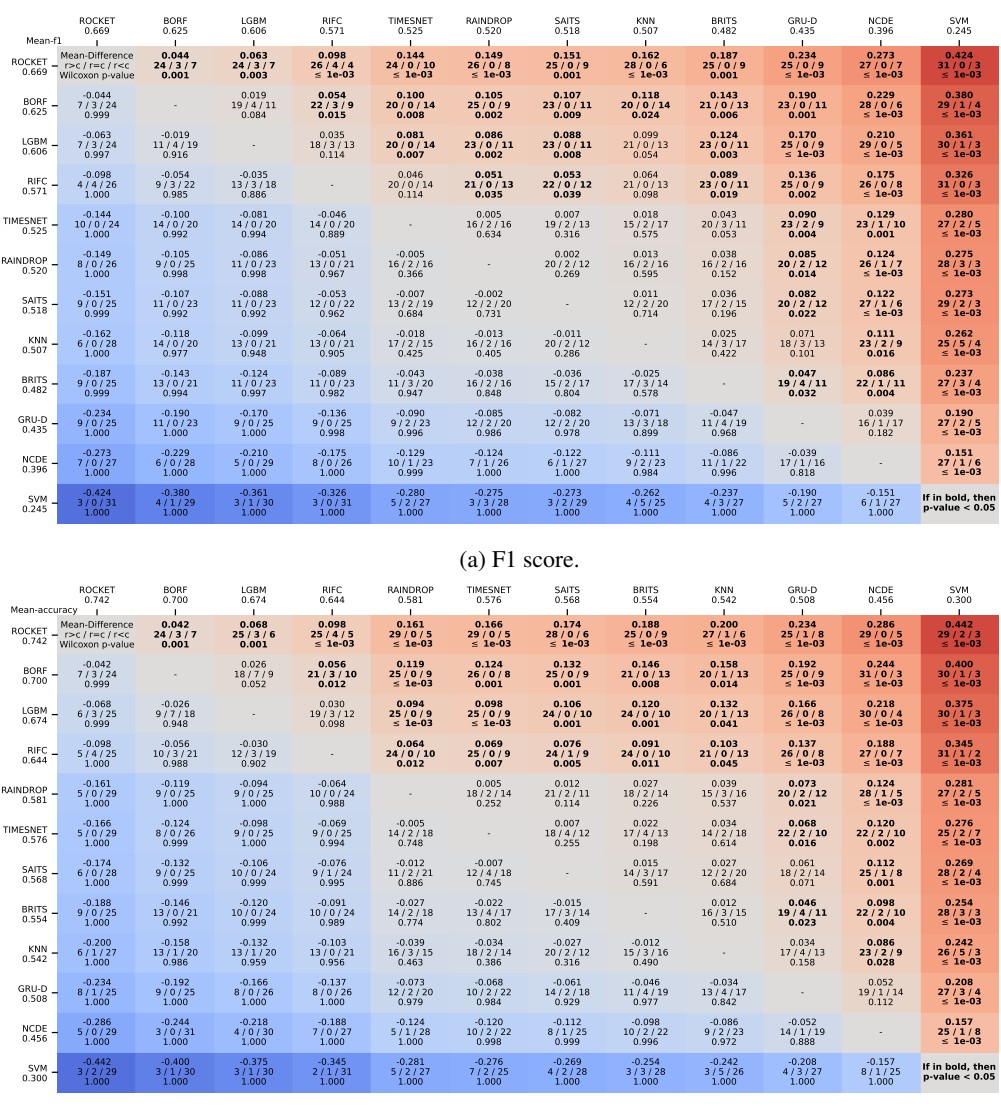

(a) F1 score.

(b) Accuracy.

Figure 12: Summary performance statistics for the 12 classifiers on 34 datasets, generated using the multiple comparison matrix (MCM). The MCM shows pairwise comparisons. Each cell shows the mean difference in performance, wins/draws/losses, and Wilcoxon p-value for two comparates. The best models on the top left are sorted based on the average performance. The more intense the color, the higher the mean accuracy difference w.r.t. the comparate, positive (red) or negative (blue).

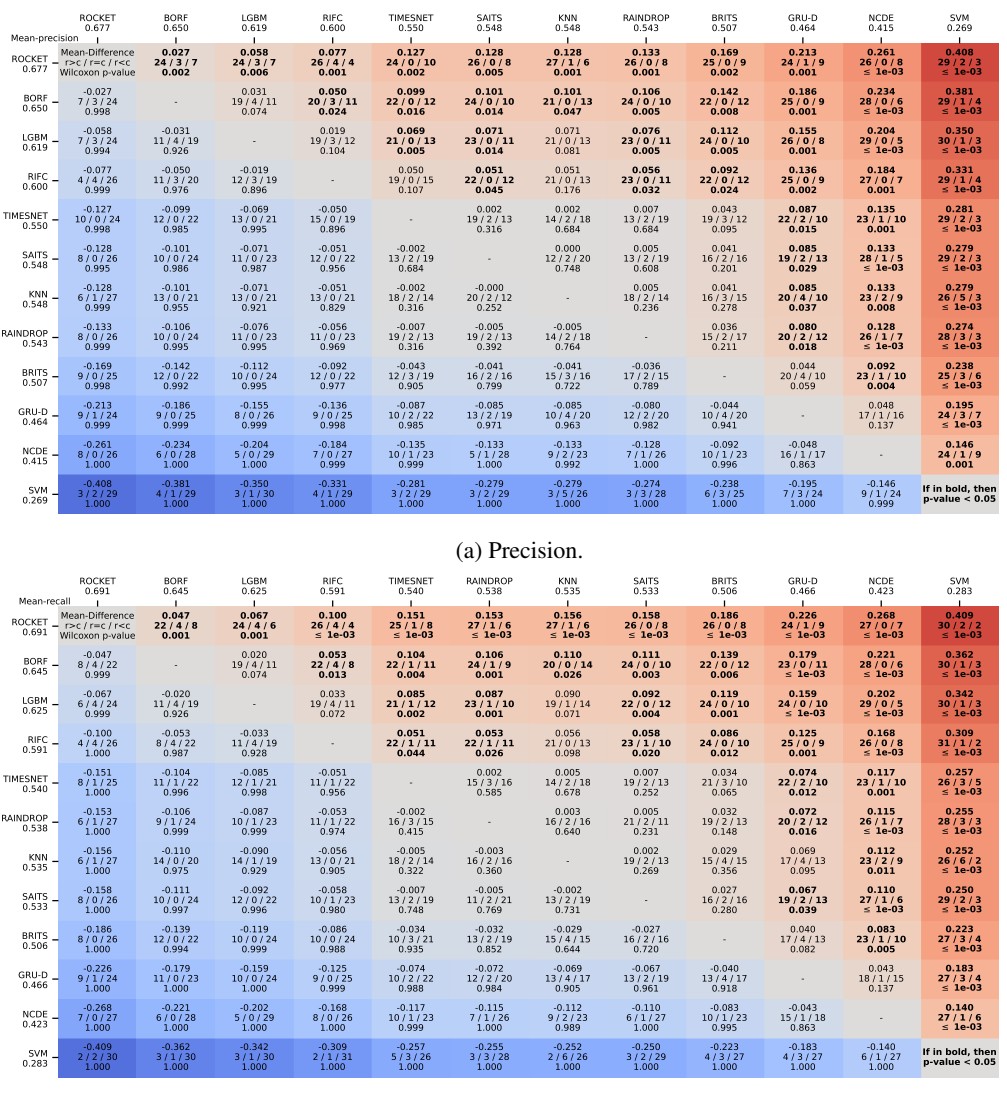

(a) Precision.

(b) Recall.

Figure 13: Summary performance statistics for the 12 classifiers on 34 datasets, generated using the multiple comparison matrix (MCM). The MCM shows pairwise comparisons. Each cell shows the mean difference in performance, wins/draws/losses, and Wilcoxon p-value for two comparates. The best models on the top left are sorted based on the average performance. The more intense the color, the higher the mean accuracy difference w.r.t. the comparate, positive (red) or negative (blue).

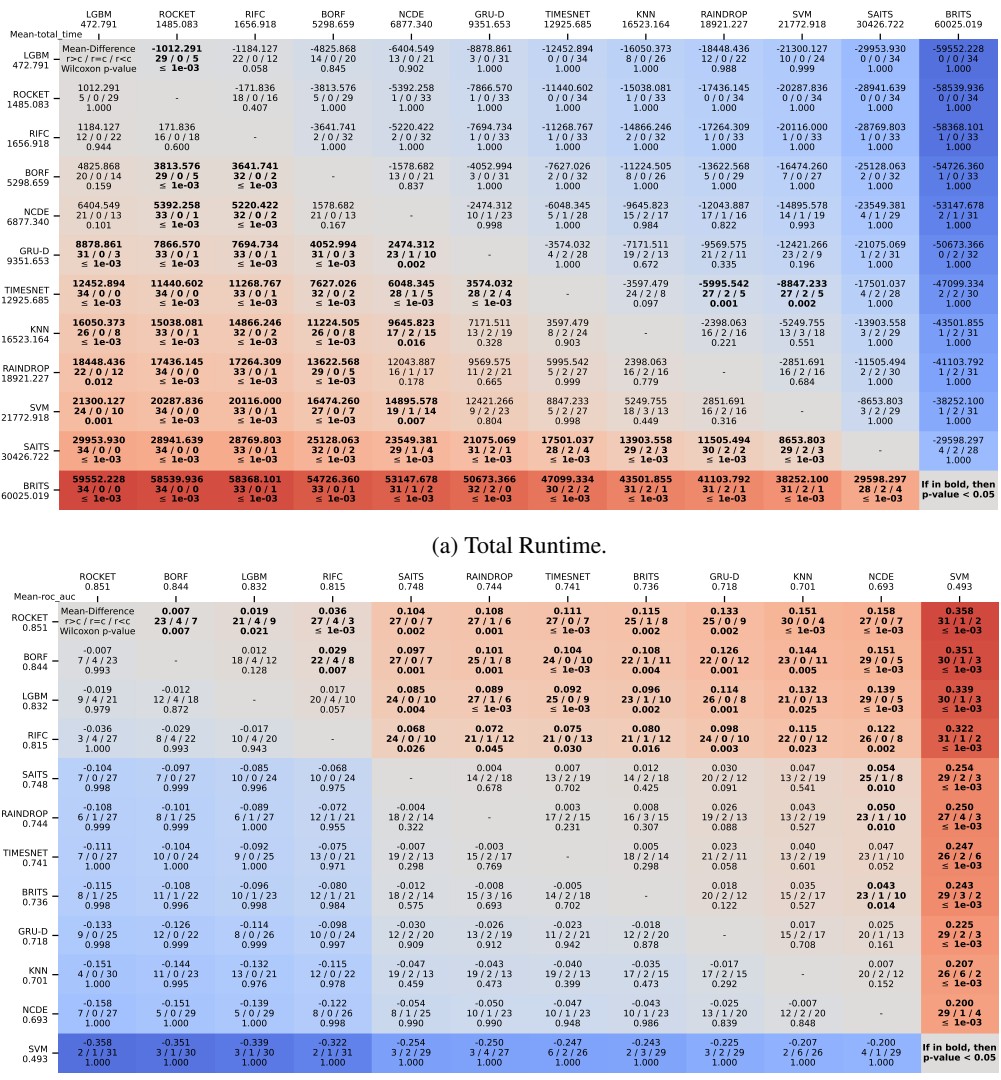

Figure 14: Summary performance statistics for the 12 classifiers on 34 datasets, generated using the multiple comparison matrix (MCM). The MCM shows pairwise comparisons. Each cell shows the mean difference in performance, wins/draws/losses, and Wilcoxon p-value for two comparates. The best models on the top left are sorted based on the average performance. The more intense the color, the higher the mean accuracy difference w.r.t. the comparate, positive (red) or negative (blue).

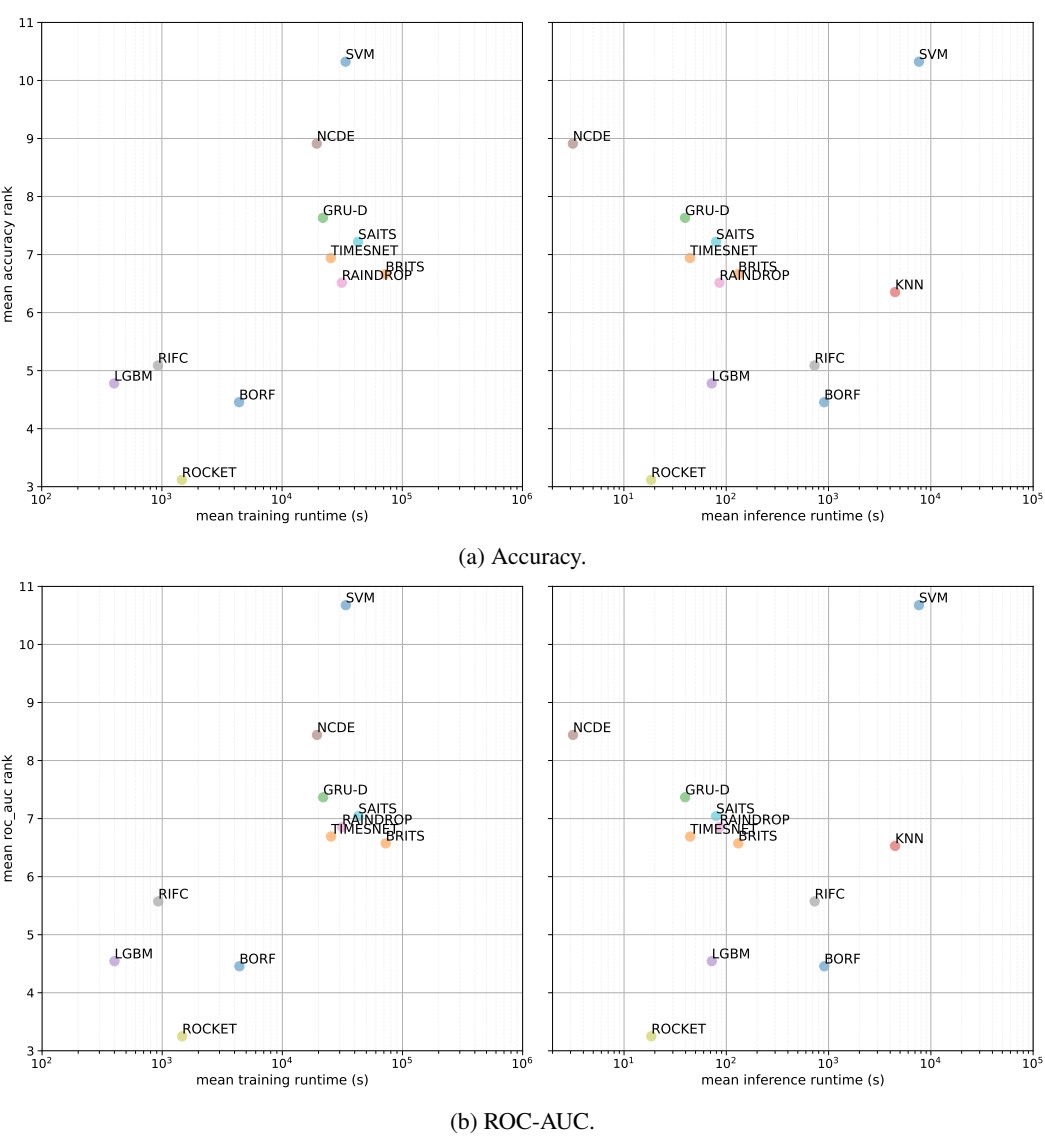

(a) Accuracy.

(b) ROC-AUC.

Figure 15: Average performance rank (lower is better) vs. training and inference runtimes (lower is better). Best values are on the bottom-left of each plot.

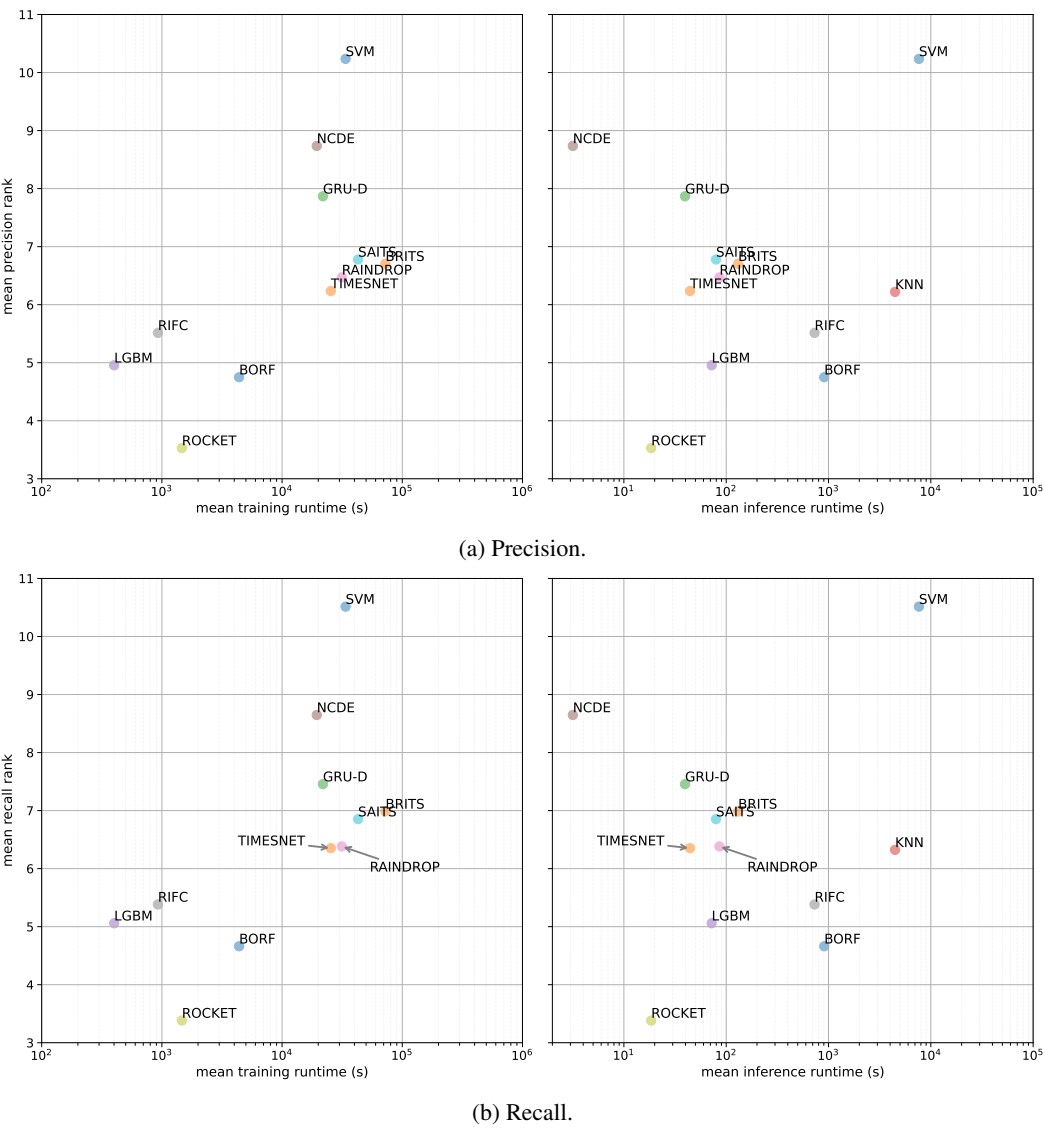

(a) Precision.

(b) Recall.

Figure 16: Average performance rank (lower is better) vs. training and inference runtimes (lower is better). Best values are on the bottom-left of each plot.

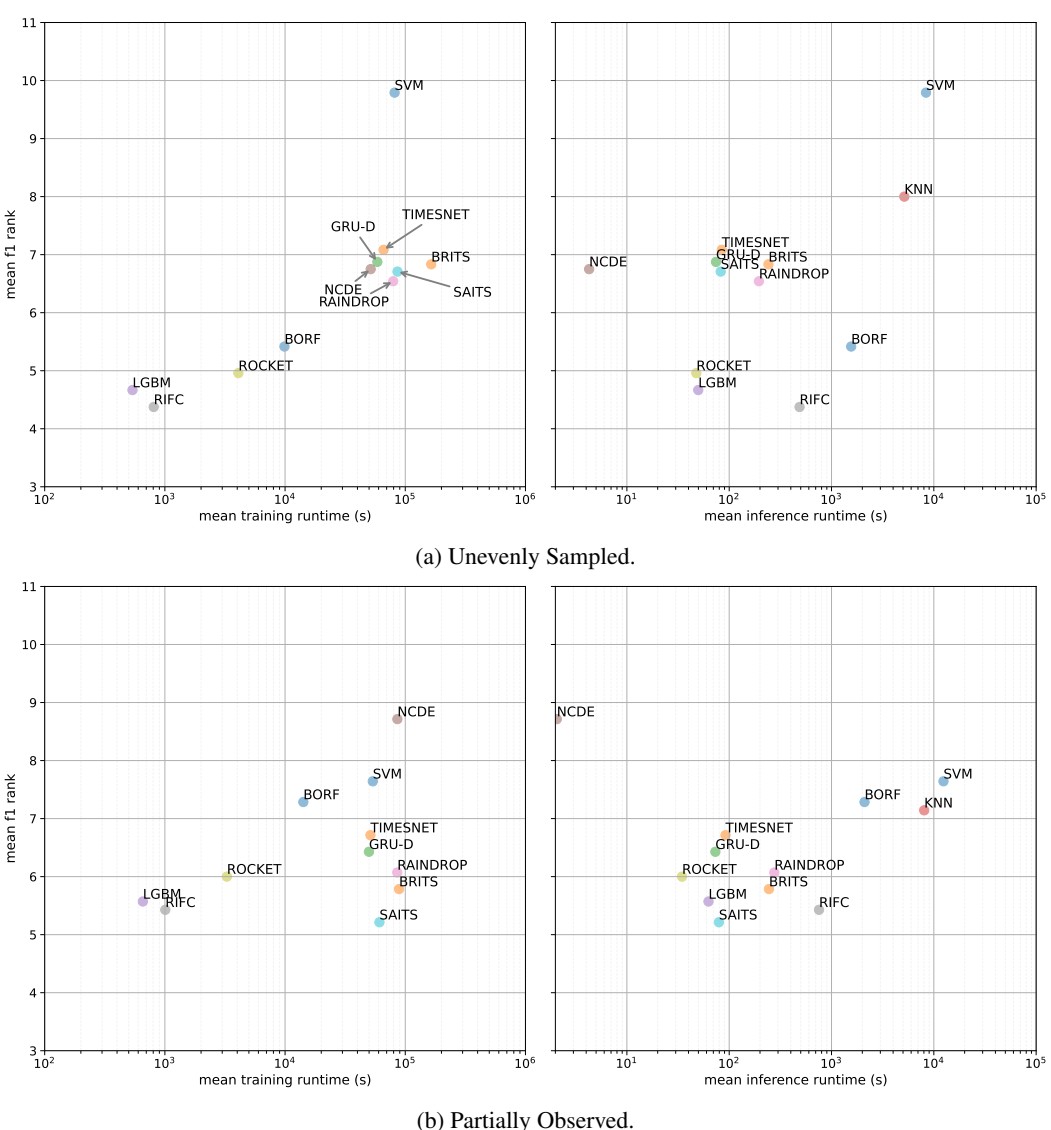

Figure 17: Average F1 rank (lower is better) vs. training and inference runtimes (lower is better) for subsets of datasets. Best values are on the bottom-left of each plot.

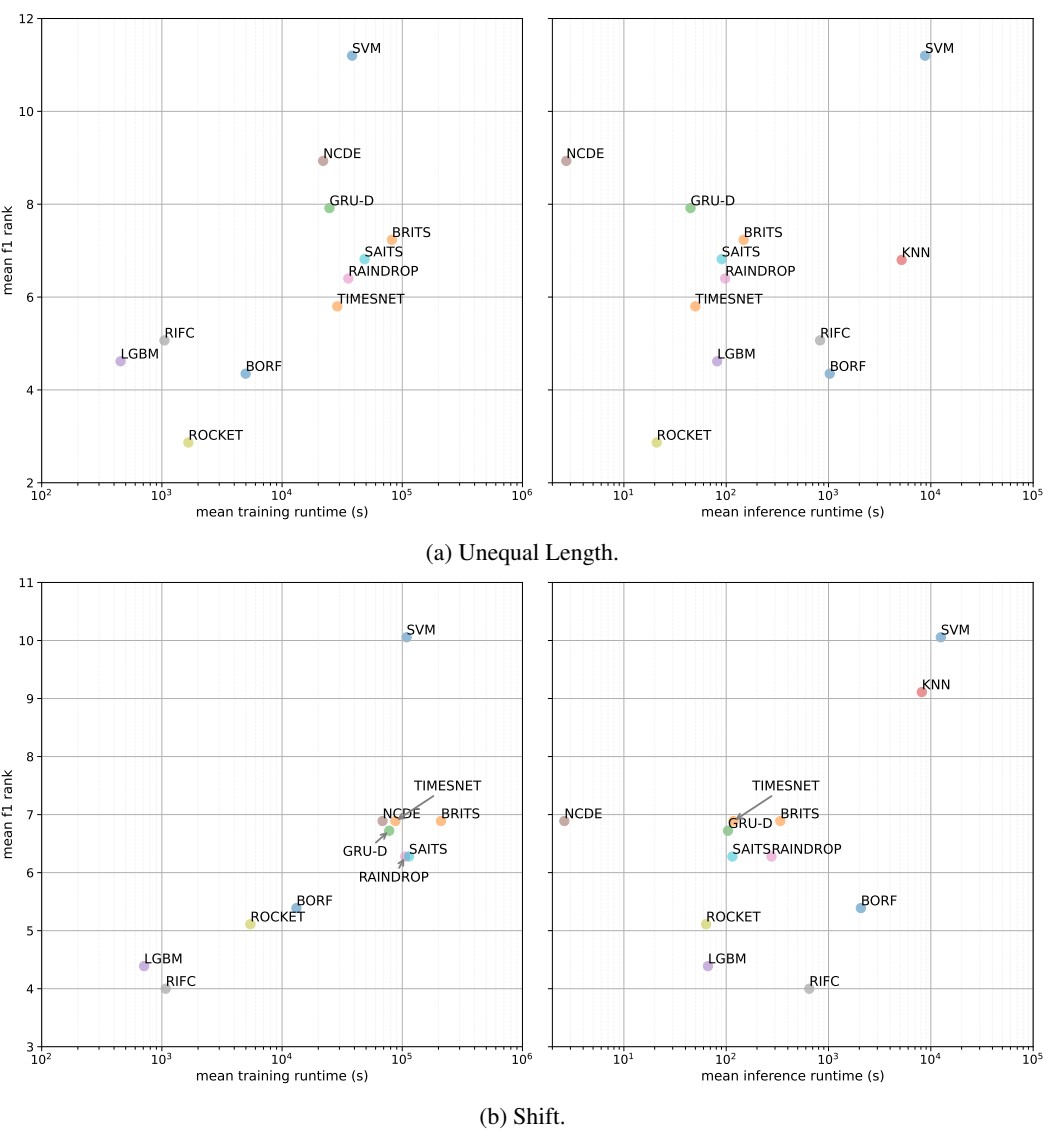

Figure 18: Average F1 rank (lower is better) vs. training and inference runtimes (lower is better) for subsets of datasets. Best values are on the bottom-left of each plot.

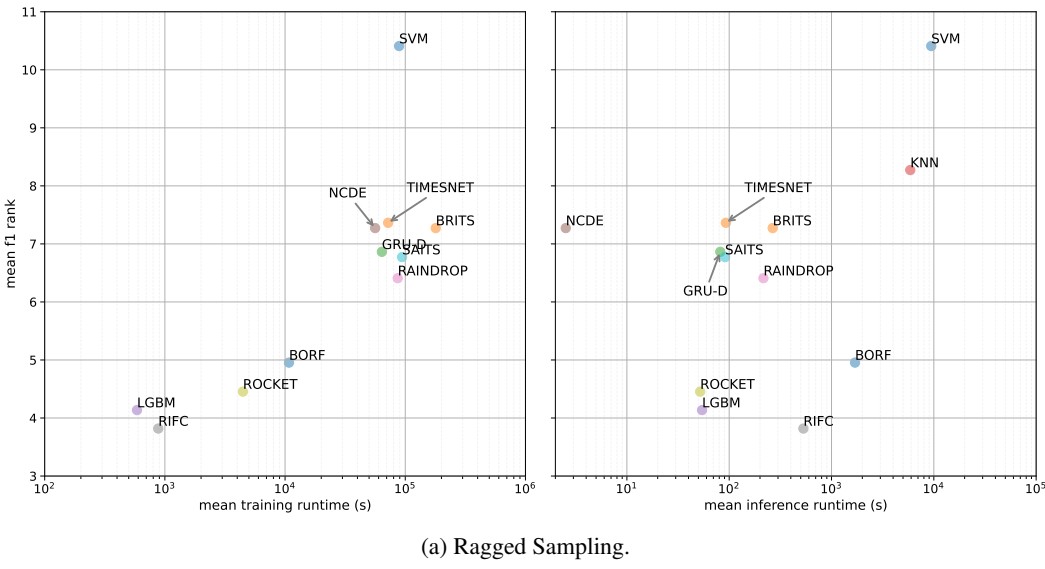

(a) Ragged Sampling.

Figure 19: Average F1 rank (lower is better) vs. training and inference runtimes (lower is better) for subsets of datasets. Best values are on the bottom-left of each plot.

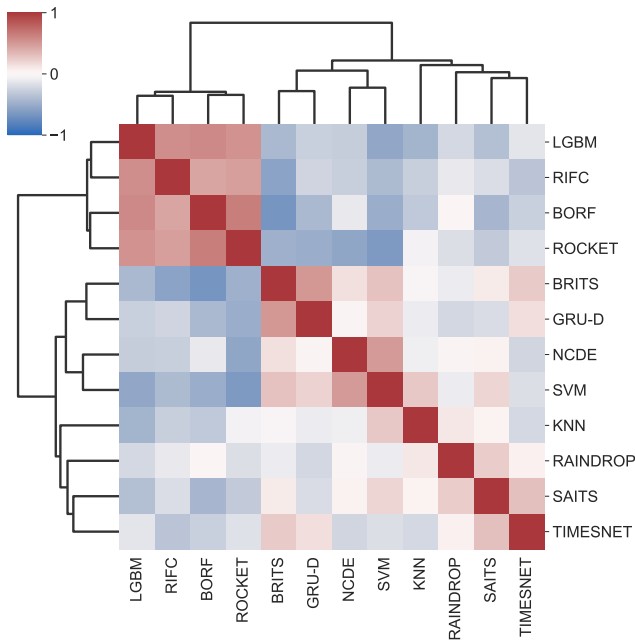

Figure 20: F1 rank correlation between models. Models are hierarchically clustered using average linkage applied to the rank correlation matrix. Positive correlations indicate that models tend to perform similarly across datasets, reflecting comparable strengths or weaknesses. Negative correlations suggest that models excel on different datasets, revealing complementary behaviors or distinct inductive biases.

Table 8: F1 score on the test set for each dataset and each classifier. The average of 3 runs is taken for methods that highly depend upon initialization, i.e., all approaches besides BORF, KNN, LGBM, and SVM. Missing values are due to exceeding memory or maximum runtime. The best values for each dataset are in bold.

| | BORF | BRITS | GRU-D | KNN | LGBM | NCDE | RAINDROP | RIFC | ROCKET | SAITS | SVM | TIMESNET |
|---|---|---|---|---|---|---|---|---|---|---|---|---|
| ABF | 0.17 | 0.33±0.01 | 0.28±0.09 | 0.31 | 0.17 | **0.41**±0.02 | 0.27±0.01 | 0.17±0.00 | 0.17±0.00 | 0.30±0.03 | 0.32 | 0.31±0.01 |
| AN | 0.80 | 0.65±0.00 | 0.68±0.03 | 0.80 | 0.80 | 0.43±0.11 | 0.64±0.05 | 0.88±0.02 | **0.90**±0.04 | 0.59±0.03 | 0.07 | 0.61±0.01 |
| AOC | **0.82** | 0.30±0.00 | 0.31±0.28 | 0.64 | 0.68 | 0.51±0.01 | 0.66±0.03 | 0.68±0.03 | 0.80±0.01 | 0.75±0.02 | 0.02 | 0.62±0.03 |
| APT | 0.91 | 0.78±0.02 | 0.49±0.31 | 0.34 | 0.80 | 0.69±0.00 | 0.77±0.01 | 0.88±0.03 | **0.96**±0.00 | 0.84±0.01 | 0.05 | 0.86±0.02 |
| ARC | 0.95 | 0.99±0.00 | 0.63±0.10 | 0.36 | 0.95 | 0.81±0.00 | 0.97±0.01 | 0.77±0.28 | **0.99**±0.01 | 0.99±0.00 | 0.10 | 0.99±0.00 |
| CT | 0.94 | 0.96±0.05 | 0.64±0.30 | **0.98** | 0.95 | 0.87±0.01 | 0.97±0.00 | 0.94±0.02 | 0.98±0.00 | 0.94±0.05 | 0.68 | 0.95±0.02 |
| DD | 0.51 | 0.52±0.02 | 0.46±0.07 | 0.44 | 0.52 | 0.29±0.12 | 0.45±0.04 | 0.49±0.04 | **0.54**±0.03 | 0.43±0.06 | 0.23 | 0.20±0.02 |
| DG | 0.34 | 0.72±0.07 | 0.71±0.07 | **0.90** | 0.34 | 0.51±0.04 | 0.60±0.22 | 0.34±0.00 | 0.34±0.00 | 0.57±0.10 | 0.85 | 0.42±0.03 |
| DW | 0.42 | 0.93±0.02 | 0.63±0.27 | **0.97** | 0.42 | 0.80±0.13 | 0.78±0.31 | 0.42±0.00 | 0.42±0.00 | 0.96±0.01 | 0.96 | 0.63±0.02 |
| GM1 | 0.58 | 0.47±0.04 | 0.24±0.08 | 0.33 | 0.57 | 0.23±0.02 | 0.46±0.11 | 0.49±0.02 | **0.66**±0.02 | 0.41±0.02 | 0.04 | 0.50±0.04 |
| GM2 | 0.50 | 0.32±0.05 | 0.40±0.08 | 0.26 | 0.39 | 0.20±0.08 | 0.30±0.15 | 0.36±0.03 | **0.57**±0.05 | 0.24±0.25 | 0.13 | 0.45±0.03 |
| GM3 | 0.34 | 0.22±0.02 | 0.06±0.02 | 0.14 | 0.25 | 0.17±0.01 | 0.31±0.03 | 0.26±0.05 | **0.48**±0.03 | 0.27±0.11 | 0.01 | 0.32±0.03 |
| GP1 | 0.88 | 0.27±0.06 | 0.23±0.14 | 0.75 | 0.78 | 0.32±0.02 | 0.81±0.05 | 0.80±0.04 | **0.89**±0.02 | 0.75±0.02 | 0.16 | 0.73±0.06 |
| GP2 | 0.79 | 0.31±0.04 | 0.56±0.24 | 0.74 | 0.73 | 0.35±0.01 | 0.63±0.10 | 0.76±0.05 | **0.85**±0.05 | 0.54±0.10 | 0.43 | 0.58±0.04 |
| GS | 0.41 | - | - | - | 0.13 | **0.52**±0.29 | - | 0.07±0.02 | 0.31±0.15 | - | - | - |
| GX | 0.55 | 0.09±0.09 | 0.11±0.04 | 0.65 | 0.50 | 0.13±0.05 | 0.44±0.08 | 0.47±0.11 | **0.70**±0.01 | 0.33±0.09 | 0.11 | 0.44±0.00 |
| GY | 0.64 | 0.18±0.07 | 0.13±0.07 | 0.65 | 0.55 | 0.26±0.01 | 0.46±0.04 | 0.49±0.14 | **0.70**±0.02 | 0.43±0.10 | 0.13 | 0.44±0.02 |
| GZ | 0.58 | 0.17±0.09 | 0.10±0.04 | 0.62 | 0.48 | 0.11±0.04 | 0.33±0.04 | 0.44±0.13 | **0.69**±0.01 | 0.19±0.12 | 0.06 | 0.33±0.02 |
| IW | 0.48 | 0.65±0.01 | 0.61±0.00 | - | **0.71** | 0.10±0.02 | 0.02±0.00 | 0.39±0.34 | 0.53±0.00 | 0.13±0.07 | 0.02 | 0.60±0.00 |
| JV | 0.71 | 0.96±0.00 | 0.96±0.01 | 0.96 | 0.93 | 0.57±0.02 | 0.94±0.02 | 0.89±0.10 | 0.94±0.01 | 0.96±0.01 | 0.47 | **0.97**±0.01 |
| LPA | **0.73** | 0.28±0.20 | 0.21±0.14 | 0.02 | 0.53 | 0.44±0.03 | 0.33±0.09 | 0.32±0.20 | 0.02±0.01 | 0.26±0.06 | 0.02 | 0.27±0.03 |
| MI3 | 0.27 | 0.42±0.11 | 0.35±0.00 | 0.35 | 0.41 | 0.34±0.17 | 0.36±0.15 | **0.56**±0.22 | 0.35±0.00 | 0.46±0.10 | 0.35 | 0.42±0.11 |
| MP | 0.85 | 0.92±0.00 | 0.92±0.01 | 0.88 | **0.96** | 0.63±0.02 | 0.74±0.04 | 0.90±0.04 | 0.94±0.00 | 0.67±0.34 | 0.44 | 0.93±0.01 |
| P12 | 0.51 | 0.46±0.00 | 0.61±0.02 | 0.12 | 0.55 | 0.49±0.01 | 0.56±0.02 | **0.63**±0.01 | 0.47±0.01 | 0.55±0.01 | 0.46 | 0.56±0.01 |
| P19 | 0.71 | 0.49±0.00 | 0.55±0.02 | - | **0.75** | - | 0.69±0.01 | 0.66±0.03 | 0.71±0.01 | 0.72±0.00 | 0.05 | 0.71±0.01 |
| PAM | 0.53 | - | - | - | 0.33 | - | - | 0.37±0.32 | **0.66**±0.10 | - | - | - |
| PGE | 0.40 | **0.78**±0.00 | **0.78**±0.00 | **0.78** | 0.40 | 0.57±0.29 | 0.48±0.26 | 0.40±0.00 | 0.40±0.00 | 0.65±0.22 | 0.40 | 0.43±0.05 |
| PGZ | 0.46 | 0.34±0.03 | 0.26±0.14 | 0.61 | 0.54 | 0.30±0.02 | 0.46±0.34 | 0.60±0.12 | **0.73**±0.00 | 0.65±0.07 | 0.16 | 0.57±0.06 |
| PL | **0.87** | 0.36±0.04 | 0.20±0.10 | 0.64 | 0.71 | 0.28±0.01 | 0.53±0.03 | 0.47±0.02 | 0.85±0.01 | 0.39±0.06 | 0.20 | 0.45±0.02 |
| SAD | 0.98 | 0.99±0.00 | 0.99±0.00 | 0.97 | 0.97 | 0.74±0.01 | 0.98±0.00 | 0.65±0.42 | 0.98±0.00 | 0.95±0.01 | 0.62 | **0.99**±0.00 |
| SE | 0.47 | 0.48±0.15 | 0.49±0.17 | 0.38 | 0.42 | 0.33±0.08 | 0.40±0.15 | **0.82**±0.04 | 0.80±0.08 | 0.27±0.02 | 0.26 | 0.24±0.06 |
| SGZ | 0.75 | 0.34±0.05 | 0.38±0.08 | 0.77 | 0.65 | 0.12±0.05 | 0.46±0.04 | 0.75±0.08 | **0.88**±0.02 | 0.57±0.11 | 0.13 | 0.49±0.09 |
| TA | 0.42 | 0.23±0.00 | 0.23±0.00 | - | **0.77** | 0.33±0.01 | 0.25±0.02 | 0.38±0.06 | 0.57±0.01 | 0.25±0.01 | - | 0.25±0.01 |
| VE | **0.97** | 0.50±0.08 | 0.60±0.01 | 0.88 | 0.94 | 0.63±0.08 | 0.65±0.02 | 0.90±0.02 | 0.94±0.02 | 0.62±0.03 | 0.40 | 0.59±0.01 |

Table 9: Total runtime (seconds) for each dataset and each classifier. The average of 3 runs is taken for methods that highly depend upon initialization, i.e., all approaches besides BORF, KNN, LGBM, and SVM. Missing values are due to exceeding memory or maximum runtime. The best values for each dataset are in bold.

| | BORF | BRITS | GRU-D | KNN | LGBM | NCDE | RAINDROP | RIFC | ROCKET | SAITS | SVM | TIMESNET |
|---|---|---|---|---|---|---|---|---|---|---|---|---|
| ABF | 16 | 230±8 | 33±6 | 10 | 2 | 1290±2051 | 16±2 | 5±0 | **1±0** | 76±16 | 20 | 65±12 |
| AN | 18 | 5952±510 | 121±24 | 8 | 12 | 93±24 | 23±4 | 2±0 | **1±0** | 344±40 | 11 | 203±29 |
| AOC | 295 | 5572±906 | 1274±503 | 1318 | 139 | 92±28 | 94±2 | **21±1** | 23±1 | 5764±903 | 1196 | 2601±438 |
| APT | 1000 | 47501±10144 | 9416±3823 | 32482 | 302 | 180±5 | 13014±21270 | 78±6 | **38±0** | 113432±7034 | 14178 | 21283±1363 |
| ARC | 679 | 246725±140403 | 16236±5490 | 26775 | 144 | 175±60 | 88376±7613 | 68±5 | **52±1** | 248253±10872 | 8021 | 18775±1290 |
| CT | 300 | 18061±4779 | 1542±719 | 2563 | 613 | 243±85 | 472±84 | **72±4** | 191±3 | 3516±590 | 2898 | 1989±292 |
| DD | 20 | 1989±478 | 90±24 | 4 | 32 | 111±29 | 39±9 | **2±0** | 7±0 | 404±84 | 22 | 232±30 |
| DG | 12 | 1063±215 | 41±6 | 3 | 1 | 111±68 | 19±9 | 1±0 | **0±0** | 174±7 | 7 | 64±12 |
| DW | 12 | 1526±904 | 67±39 | 3 | 1 | 243±141 | 40±37 | 1±0 | **0±0** | 223±103 | 7 | 310±378 |
| GM1 | 18 | 10400±2849 | 477±163 | 95 | 346 | 158±42 | 76±19 | **4±0** | 25±1 | 1156±169 | 102 | 586±104 |
| GM2 | 20 | 15746±1939 | 466±86 | 94 | 364 | 188±100 | 156±37 | **4±0** | 26±1 | 1528±459 | 103 | 1078±266 |
| GM3 | 22 | 7442±1118 | 467±222 | 93 | 440 | 136±39 | 80±18 | **4±0** | 28±1 | 1228±122 | 103 | 634±26 |
| GP1 | 25 | 6802±5500 | 338±14 | 82 | 61 | 91±27 | 56±9 | **3±0** | 5±0 | 1450±360 | 97 | 575±165 |
| GP2 | 27 | 6579±1563 | 998±53 | 79 | 71 | 90±25 | 50±9 | **3±0** | 5±0 | 1253±78 | 105 | 797±149 |
| GS | 5718 | - | - | - | **1358** | 4815±6530 | - | 1826±22 | 26010±136 | - | - | - |
| GX | 62 | 4846±1652 | 430±133 | 942 | 373 | 52±25 | 85±5 | **7±1** | 17±1 | 2015±341 | 307 | 1202±373 |
| GY | 54 | 5314±447 | 612±77 | 939 | 374 | 59±11 | 78±2 | **7±0** | 17±0 | 2519±362 | 311 | 1151±198 |
| GZ | 63 | 6439±3059 | 533±67 | 879 | 230 | 87±24 | 97±3 | **7±0** | 17±1 | 2864±647 | 306 | 1194±225 |
| IW | 31587 | 7067±138 | 1460±62 | - | 4533 | 38767±8867 | 5937±1596 | 39775±851 | **118±2** | 5257±248 | 279477 | 5825±1060 |
| JV | 23 | 289±79 | 51±0 | 48 | 133 | 92±27 | 89±7 | 35±2 | **6±0** | 197±112 | 41 | 337±42 |
| LPA | 411 | 31534±11204 | 3854±303 | 447 | 278 | 186±10 | 10318±17149 | 46±1 | **27±0** | 39939±5076 | 99 | 35634±26815 |
| MI3 | 12 | 1421±335 | 41±10 | 3 | 3 | 76±16 | 14±2 | 6±0 | **0±0** | 113±35 | 4 | 81±12 |
| MP | 26 | 6230±7 | 122±13 | 816 | 306 | 205±85 | 550±207 | **18±1** | 142±1 | 1171±241 | 950 | 489±90 |
| P12 | 5455 | 72508±32359 | 4182±948 | 40226 | 239 | 1204±205 | 7582±12025 | 1907±34 | **22±1** | 9444±1250 | 5764 | 78817±1145 |
| P19 | 30858 | 244447±112157 | 45315±6110 | - | 751 | - | 294468±72194 | 9243±919 | **206±2** | 116971±22038 | 144163 | 51343±6196 |
| PA2 | 77647 | - | - | - | 4004 | - | - | **1179±97** | 22963±92 | - | - | - |
| PGE | 6 | 784±165 | 50±69 | 2 | 1 | 197±123 | 9±1 | 2±0 | **0±0** | 39±25 | 2 | 21±3 |
| PGZ | 8 | 2182±670 | 245±142 | 11 | **1** | 110±28 | 30±15 | 1±0 | 1±0 | 365±63 | 8 | 197±15 |
| PL | 108 | 56920±23275 | 4615±3922 | 4017 | 452 | 106±2 | 5162±8392 | **11±0** | 40±1 | 26156±2239 | 2666 | 7031±1167 |
| SAD | 8487 | 33021±5648 | 1829±133 | 17309 | **82** | 930±143 | 1966±398 | 675±53 | 103±1 | 6956±536 | 17106 | 6991±936 |
| SE | 204 | 80101±34115 | 1893±120 | 164 | 17 | 123±31 | 20918±2925 | **9±0** | 39±1 | 59800±14909 | 2019 | 3122±594 |
| SGZ | 9 | 1821±119 | 267±29 | 12 | 28 | 115±69 | 25±5 | **1±0** | 2±0 | 325±20 | 10 | 239±34 |
| TA | 16825 | 861475±86400 | 46781±20524 | - | 352 | 10592±6509 | 20576±7435 | 1305±19 | **350±9** | 197788±30354 | - | 92470±29583 |
| VE | 127 | 76064±5836 | 1311±589 | 362 | 32 | 115±49 | 108±27 | 8±0 | **6±0** | 10987±6396 | 976 | 2339±196 |

# E    ARRAY STRUCTURES

We report a summary of the main formats used to represent regular and irregular time series data in the literature in Table 10.

Table 10: Overview of the main formats used to represent regular and irregular time series data in the literature, categorized by tensor type. The table details the underlying data structures (classes), the software libraries that implement them, their usage across the time series libraries considered in this study, and their support for timestamps and tensor operations.

| Type | Format | Library | Class | Usage | Timestamps | Tensor Ops. |
|---|---|---|---|---|---|---|
| Dense | 3D Tensor | numpy | Array | aeon | ✗ | ✓ |
| | | numpy | Array | sktime | ✗ | ✓ |
| | | numpy | Array | tslearn | ✗ | ✓ |
| | | numpy | MaskedArray | - | ✗ | ✓ |
| | | jax | Array | diffrax | ✓* | ✓ |
| | | tensorflow | Array | - | ✗ | ✓ |
| | | torch | Tensor | pypots | ✗ | ✓ |
| Ragged | 3D Tensor | awkward | AwkwardArray | - | ✗ | ✓ |
| | | tensorflow | RaggedTensor | - | ✗ | ✓ |
| | | torch | NestedTensor | - | ✗ | ✓ |
| | | zarr | RaggedArray | - | ✗ | ✓ |
| | | pyarrow | ListArray | - | ✗ | ✓ |
| Sparse | 3D Tensor | sparse | GCXS | - | ✗ | ✓ |
| | | sparse | DOK | - | ✗ | ✓ |
| | | sparse | COO | - | ✗ | ✓ |
| Other | Nested List | python | List[Array] | aeon | ✗ | ✗ |
| | 3D tensor** | xarray | Dataset | - | ✓ | ✓ |
| | Long | pandas | DataFrame | sktime | ✓ | ✗ |
| | MultiIndex | pandas | DataFrame | sktime | ✓ | ✗ |

\* only as a separate channel
\*\* with additional tensors for static variables

# F    EXTENDING PYRREGULAR TO OTHER TASKS

As noted in Section 6, our framework is designed to extend naturally to several additional tasks beyond classification. In particular, we highlight regression, forecasting, and anomaly detection, which are already supported at the representation level and require only minor adjustments to dataset metadata or the inclusion of auxiliary variables.

**Regression.** This task involves predicting continuous outcomes and is directly supported by our framework. Typical targets include SAPS-I (Simplified Acute Physiology Score) in PhysioNet 2012 or raw productivity in the Garment dataset. To demonstrate feasibility, we provide an early, illustrative benchmark in Table 11. It covers four datasets (MI3, P12, P19, PGE) and evaluates the top three generalist methods from the classification experiments (changing the head to a regression one), alongside a NAIVE baseline that returns the mean target value. No tuning was performed, and the target definitions have not yet been curated; this example is intended only as an initial proof of concept. A preliminary code snippet for regression is included in Appendix G.

Table 11: Pilot regression benchmark in terms of *rmse* on a selected number of datasets and classifiers. Lower is better, best results in bold.

|       | *target*  | NAIVE | BORF  | LGBM      | ROCKET |
|-------|-----------|-------|-------|-----------|--------|
| MI3   | *Age*       | 20.55 | 21.44 | **16.67** | 20.21  |
| P12   | *SAPS-I*    | 5.97  | 4.35  | **4.11**  | 5.67   |
| P19   | *Age*       | 16.79 | 16.56 | **14.62** | 16.68  |
| PGE   | *Raw Prod.* | 0.09  | 0.09  | 0.09      | 0.09   |

**Forecasting.** Here the objective is to predict future values of a time series given its history. We plan to introduce a static variable with a cutoff point to indicate the train/test split, and to extend the accessor method to provide users with a straightforward mechanism for performing this split.

**Anomaly detection.** This task aims to identify unusual or irregular patterns in the data. Since anomalies may have the same shape as the underlying dataset, they cannot be indicated via static variables. Instead, leveraging the support for additional data arrays in `xarray`, we will represent anomalies using sparse binary masks that flag anomalous regions in the time series.

**Model support.** We also plan to support a set of representative models for such tasks. A non-comprehensive list includes recent work introducing dynamic graph networks for medical data (Luo et al., 2024), image-based transformers for irregular series (Li et al., 2023), channel harmony strategies (Liu et al., 2025a), graph neural flows (Mercatali et al., 2024), temporal graph ODEs (Gravina et al., 2024), state space models (Gu et al., 2022), patching graph neural networks for forecasting (Zhang et al., 2024), and TabPFN with feature engineering (Hollmann et al., 2023) to name a few. We further aim to incorporate (at least from an inference perspective) emerging pre-trained generalist frameworks designed to operate across diverse temporal domains. These include large-language-model-based temporal reasoning architectures (e.g., Time-LLM, (Jin et al., 2024)), cross-modal alignment frameworks such as CALF (Liu et al., 2025b), and multimodal pretraining strategies capable of unifying sequential signals from heterogeneous sources (King et al., 2023). Such models highlight a trend toward universal temporal representations that can handle varying sampling rates, modalities, and tasks within a single foundation model.

# G    QUICK GUIDE

Extensive documentation and examples are available at `https://github.com/fspinna/pyrregular`. Below, we provide a quick start guide and simple workflow notebooks.

```
pip install pyrregular[models]
```

## G.1    LIST DATASETS

If you want to see all the datasets available, you can use the `list_datasets` function:

```python
from pyrregular import list_datasets
df = list_datasets()
```

## G.2    LOAD A DATASET

To load a dataset, you can use the `load_dataset` function. For example, to load the "Garment" dataset, you can do:

```python
from pyrregular import load_dataset
df = load_dataset("Garment.h5")
```

## G.3    CLASSIFICATION

To use the dataset for classification, you can just "densify" it:

```python
from pyrregular import load_dataset

df = load_dataset("Garment.h5")
X, _ = df.irr.to_dense()
y, split = df.irr.get_task_target_and_split()

X_train, X_test = X[split != "test"], X[split == "test"]
y_train, y_test = y[split != "test"], y[split == "test"]

# We have ready-to-go models from various libraries:
from pyrregular.models.rocket import rocket_pipeline

model = rocket_pipeline
model.fit(X_train, y_train)
model.score(X_test, y_test)
```

The dataset can be also easily used in `pytorch`

```python
from torch.utils.data import DataLoader, TensorDataset
import torch

data = TensorDataset(X, y)
dataloader = DataLoader(data, batch_size=16, shuffle=True)
```

## Notebook: Basic Workflow

```python
import pandas as pd
import xarray as xr
```

### List available datasets

To view available datasets, you can use the `list_datasets` function.

```python
from pyrregular import list_datasets
```

```python
print(list_datasets())
```

```
['Abf.h5', 'AllGestureWiimoteX.h5', 'AllGestureWiimoteY.h5',
'AllGestureWiimoteZ.h5', 'Animals.h5', 'AsphaltObstaclesCoordinates.h5',
'AsphaltPavementTypeCoordinates.h5', 'AsphaltRegularityCoordinates.h5',
'CharacterTrajectories.h5', 'DodgerLoopDay.h5',
'DodgerLoopGame.h5', 'DodgerLoopWeekend.h5', 'Garment.h5',
'GeolifeSupervised.h5', 'GestureMidAirD1.h5', 'GestureMidAirD2.h5',
'GestureMidAirD3.h5', 'GesturePebbleZ1.h5', 'GesturePebbleZ2.h5',
'JapaneseVowels.h5', 'Ldfpa.h5', 'MelbournePedestrian.h5', 'Mimic3.h5',
'PLAID.h5', 'Pamap2.h5', 'Physionet2012.h5', 'Physionet2019.h5',
'PickupGestureWiimoteZ.h5', 'Seabirds.h5', 'ShakeGestureWiimoteZ.h5',
'SpokenArabicDigits.h5', 'Taxi.h5', 'Vehicles.h5']
```

### Loading the dataset from the online repository

Loading a dataset is as from the online repo is as simple as calling the `load_dataset` function with the dataset name.

```python
from pyrregular import load_dataset
```

```python
ds = load_dataset("Garment.h5")
```

The dataset is loaded as an xarray dataset. The dataset is saved in the default os cache directory, which can be found with:

```python
import pooch
print(pooch.os_cache("pyrregular"))
```

You can also use xarray to directly load a local file. In this case, you have to specify our backend as pyrregular in the `engine` argument.

```python
import xarray as xr
ds = xr.load_dataset("path/to/file.h5", engine="pyrregular")
```

You can view the underlying DataArray by calling the `data` variable.

```python
da = ds.data
```

```python
da
```

```
[Out]: <xarray.DataArray 'data' (ts_id: 24, signal_id: 9, time_id: 59)> Size: 329kB
       <COO: shape=(24, 9, 59), dtype=float64, nnz=10267, fill_value=nan>
       Coordinates:
           day                   (time_id) <U9 2kB 'Thursday' ... 'Wednesday'
           department            (ts_id) <U9 864B 'finishing' ... 'swing'
           productivity_binary   (ts_id) int32 96B 1 0 1 1 1 1 1 1 ... 1 1 0 0 0 0 1
           productivity_class    (ts_id) <U4 384B 'high' 'low' ... 'low' 'high'
           productivity_numerical (ts_id) float32 96B 0.8126 0.6283 ... 0.7005 0.7503
           quarter               (time_id) <U8 2kB 'Quarter1' ... 'Quarter2'
         * signal_id             (signal_id) <U21 756B 'idle_men' ... 'wip'
           split                 (ts_id) <U5 480B 'train' 'train' ... 'train' 'train'
           team                  (ts_id) int32 96B 1 10 11 12 2 3 4 ... 3 4 5 6 7 8 9
         * time_id               (time_id) datetime64[ns] 472B 2015-01-01T01:00:00...
         * ts_id                 (ts_id) <U12 1kB 'finishing_1' ... 'swing_9'
       Attributes:
           _fixed_at:  2024-12-04T21:50:44.408790-12:00
           _is_fixed:  True
           author:     [Abdullah Al Imran, Md Shamsur Rahim, Tanvir Ahmed]
           configs:    {'default': {'task': 'classification', 'split': 'split', 'tar...
           license:    CC BY 4.0
           source:     https://archive.ics.uci.edu/dataset/597/productivity+predicti...
           title:      Productivity Prediction of Garment Employees
```

```python
[In]: # the shape is (n_time_series, n_channels, n_timestamps)
      da.shape
```

```
[Out]: (24, 9, 59)
```

```python
[In]: # the array is stored as a sparse array
      da.data
```

```
[Out]: <COO: shape=(24, 9, 59), dtype=float64, nnz=10267, fill_value=nan>
```

```python
[In]: # dimensions contain the time series ids, signal ids and timestamps
      da.dims
```

```
[Out]: ('ts_id', 'signal_id', 'time_id')
```

```python
[In]: # e.g., these are the time series ids
      da["ts_id"].data
```

```
[Out]: array(['finishing_1', 'finishing_10', 'finishing_11', 'finishing_12',
              'finishing_2', 'finishing_3', 'finishing_4', 'finishing_5',
              'finishing_6', 'finishing_7', 'finishing_8', 'finishing_9',
              'swing_1', 'swing_10', 'swing_11', 'swing_12', 'swing_2',
              'swing_3', 'swing_4', 'swing_5', 'swing_6', 'swing_7',
              'swing_8', 'swing_9'], dtype='<U12')
```

```
[In]: # there are also static variables, such as the class
      da["productivity_binary"].data
```

```
[Out]: array([1, 0, 1, 1, 1, 1, 1, 1, 0, 0, 0, 0, 1, 1, 1, 1, 1, 1, 1, 0, 0, 0,
              0, 1], dtype=int32)
```

```
[In]: # the train/test split
      da["split"].data
```

```
[Out]: array(['train', 'train', 'test', 'train', 'train', 'test', 'train',
              'train', 'train', 'test', 'train', 'train', 'test', 'train',
              'train', 'test', 'train', 'train', 'train', 'train', 'test',
              'train', 'train', 'train'], dtype='<U5')
```

```
[In]: # all the coordinates can be accessed via the `coords` variable
      da.coords
```

```
[Out]: Coordinates:
          day                  (time_id) <U9 2kB 'Thursday' ... 'Wednesday'
          department           (ts_id) <U9 864B 'finishing' ... 'sweing'
          productivity_binary  (ts_id) int32 96B 1 0 1 1 1 1 1 1 ... 1 1 0 0 0 0 1
          productivity_class   (ts_id) <U4 384B 'high' 'low' ... 'low' 'high'
          productivity_numerical  (ts_id) float32 96B 0.8126 0.6283 ... 0.7005 0.7503
          quarter              (time_id) <U8 2kB 'Quarter1' ... 'Quarter2'
        * signal_id            (signal_id) <U21 756B 'idle_men' ... 'wip'
          split                (ts_id) <U5 480B 'train' 'train' ... 'train' 'train'
          team                 (ts_id) int32 96B 1 10 11 12 2 3 4 ... 3 4 5 6 7 8 9
        * time_id              (time_id) datetime64[ns] 472B 2015-01-01T01:00:00...
        * ts_id                (ts_id) <U12 1kB 'finishing_1' ... 'sweing_9'
```

```
[In]: # metadata contains informations about the datasets and tasks
      da.attrs
```

```
[Out]: {'_fixed_at': '2024-12-04T21:50:44.408790-12:00',
        '_is_fixed': True,
        'author': ['NA'],
        'configs': {'default': {'task': 'classification',
          'split': 'split',
          'target': 'productivity_binary'},
         'regression': {'task': 'regression',
          'split': 'split',
          'target': 'productivity_numerical'}},
        'license': 'CC BY 4.0',
        'source': 'https://archive.ics.uci.edu/dataset/597/productivity+prediction+of+g
       arment+employees',
        'title': 'Productivity Prediction of Garment Employees'}
```

### Data Handling and Plotting

Data can be accessed with standard xarray methods.

```
[In]:  import matplotlib.pyplot as plt
       import numpy as np
```

```
[In]:  # the first time series
       da[0]
```

```
[Out]: <xarray.DataArray 'data' (signal_id: 9, time_id: 59)> Size: 9kB
       <COO: shape=(9, 59), dtype=float64, nnz=392, fill_value=nan>
       Coordinates:
           day                    (time_id) <U9 2kB 'Thursday' ... 'Wednesday'
           department             <U9 36B 'finishing'
           productivity_binary    int32 4B 1
           productivity_class     <U4 16B 'high'
           productivity_numerical float32 4B 0.8126
           quarter                (time_id) <U8 2kB 'Quarter1' ... 'Quarter2'
         * signal_id              (signal_id) <U21 756B 'idle_men' ... 'wip'
           split                  <U5 20B 'train'
           team                   int32 4B 1
         * time_id                (time_id) datetime64[ns] 472B 2015-01-01T01:00:00...
           ts_id                  <U12 48B 'finishing_1'
       Attributes:
           _fixed_at:  2024-12-04T21:50:44.408790-12:00
           _is_fixed:  True
           author:     ['NA']
           configs:    {'default': {'task': 'classification', 'split': 'split', 'tar...
           license:    CC BY 4.0
           source:     https://archive.ics.uci.edu/dataset/597/productivity+predicti...
           title:      Productivity Prediction of Garment Employees
```

```
[In]:  # the first channel of the first time series
       da[0, 0]
```

```
[Out]: <xarray.DataArray 'data' (time_id: 59)> Size: 784B
       <COO: shape=(59,), dtype=float64, nnz=49, fill_value=nan>
       Coordinates:
           day                    (time_id) <U9 2kB 'Thursday' ... 'Wednesday'
           department             <U9 36B 'finishing'
           productivity_binary    int32 4B 1
           productivity_class     <U4 16B 'high'
           productivity_numerical float32 4B 0.8126
           quarter                (time_id) <U8 2kB 'Quarter1' ... 'Quarter2'
           signal_id              <U21 84B 'idle_men'
           split                  <U5 20B 'train'
           team                   int32 4B 1
```

```
      * time_id                     (time_id) datetime64[ns] 472B 2015-01-01T01:00:00...
        ts_id                       <U12 48B 'finishing_1'
    Attributes:
        _fixed_at:    2024-12-04T21:50:44.408790-12:00
        _is_fixed:    True
        author:       ['NA']
        configs:      {'default': {'task': 'classification', 'split': 'split', 'tar...
        license:      CC BY 4.0
        source:       https://archive.ics.uci.edu/dataset/597/productivity+predicti...
        title:        Productivity Prediction of Garment Employees
```

[In]: *# to access the underlying sparse vector*
```
da[0, 0].data
```

[Out]: `<COO: shape=(59,), dtype=float64, nnz=49, fill_value=nan>`

[In]: *# to access the underlying dense vector*
```
da[0, 4].data.todense()
```

[Out]:
```
array([ 8.,  8.,  8.,  8.,  8.,  8.,  8.,  8.,  8.,  8.,  2.,  8.,  8.,
        8., nan, nan, nan,  8., 25.,  8.,  8., 10., 10., 10., 10., 15.,
       19., 19., 10., 10., 12., 10., 10., 10., 12., 12., 12., 12.,  8.,
       nan, nan, nan, nan, 12., nan, nan, nan,  8.,  8.,  8.,  8.,  8.,
        8.,  8.,  8.,  8.,  8.,  8.,  8.])
```

[In]: *# this vector contains a lot of nans, which are the padding necessary to have*␣
    ↪*shared timestamps w.r.t. the whole dataset*
```
np.isnan(da[0, 4].data.todense()).sum()
```

[Out]: 10

[In]:
```
plt.plot(da[0, 4]["time_id"], da[0, 4], marker="o")
```

[Out]: `[<matplotlib.lines.Line2D at 0x172a35b50>]`

```
[In]: # using the custom ".irr" accessor, we can filter out the nans to the minimum␣
      ↪amount possible due to raggedness
      np.isnan(da.irr[0, 4].data.todense()).sum()
```

[Out]: 0

```
[In]: plt.plot(da.irr[0, 4]["time_id"], da.irr[0, 4], marker="o")
```

[Out]: [<matplotlib.lines.Line2D at 0x172afec30>]

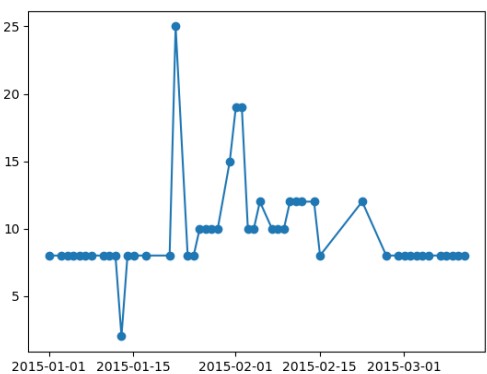

```
[In]: # the fourth channel first 10 time series of the dataset, as a heatmap
      da.irr[:10, 4].plot()
```

[Out]: <matplotlib.collections.QuadMesh at 0x172aefe90>

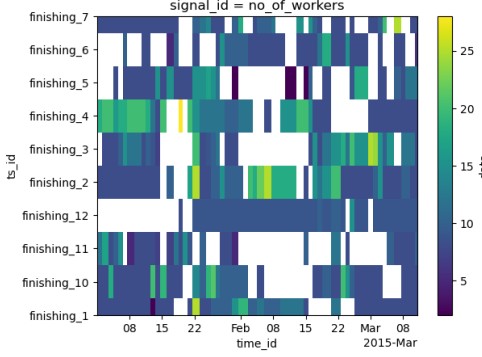

```

```
[In]:  # plotting some channels
       da.irr[0, 2].plot(label=da.coords["signal_id"][2].item())
       da.irr[0, 4].plot(label=da.coords["signal_id"][4].item())
       da.irr[0, 5].plot(label=da.coords["signal_id"][5].item())
       plt.legend()
```

[Out]:  <matplotlib.legend.Legend at 0x30fe61be0>

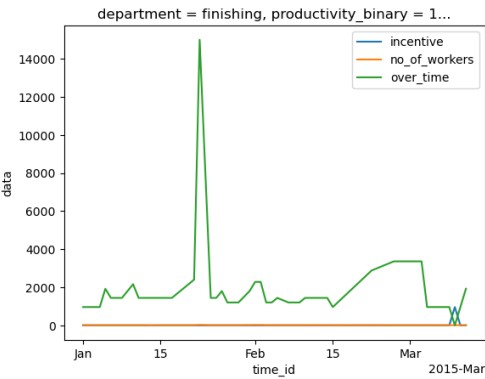

### Downstream Tasks

The xarray is nice, but not supported by basically any downstream library. Thus, we can convert it into a numpy array.

```
[In]:  %%time
       # time series data, timestamps
       X, T = da.irr.to_dense(
           normalize_time=True,  # normalize the time index to [0, 1]
       )
```

```
CPU times: user 14.9 ms, sys: 1.57 ms, total: 16.4 ms
Wall time: 5.14 ms
```

```
[In]:  # the shape is (n_time_series, n_channels, n_timestamps), timestamps are␣
       ↪returned as a separate channel, for downstream methods that are able to use␣
       ↪them
       X.shape, T.shape
```

[Out]:  ((24, 9, 59), (24, 1, 59))

```
[In]: # static variables
      Z = da.coords.to_dataset()[["split", "productivity_binary"]].to_pandas()
      Z.head()
```

```
[Out]:              split  productivity_binary department productivity_class  \
      ts_id
      finishing_1   train                    1  finishing               high
      finishing_10  train                    0  finishing                low
      finishing_11   test                    1  finishing               high
      finishing_12  train                    1  finishing               high
      finishing_2   train                    1  finishing               high

                    productivity_numerical  team
      ts_id
      finishing_1                 0.812625     1
      finishing_10                0.628333    10
      finishing_11                0.874028    11
      finishing_12                0.922840    12
      finishing_2                 0.819271     2
```

```
[In]: # target and split
      y, split = da.irr.get_task_target_and_split()
```

**Train-test split**

```
[In]: X_train, X_test = X[split != "test"], X[split == "test"]
      y_train, y_test = y[split != "test"], y[split == "test"]
      X_train.shape, y_train.shape, X_test.shape, y_test.shape
```

```
[Out]: ((18, 9, 59), (18,), (6, 9, 59), (6,))
```

**Classification**

We have several ready-to-use classifiers in the `pyrregular` package. Be sure to install the required dependencies.

```
[In]: from pyrregular.models.rocket import rocket_pipeline
```

```
[In]: %%time
      model = rocket_pipeline
      model.fit(X_train, y_train)
      model.score(X_test, y_test)
```

```
[Out]: 0.6666666666666666
```

**Regression (work in progress)**

Some datasets have regression tasks defined. The usage is the same as for classification.

```
[In]: from sklearn.metrics import mean_absolute_percentage_error
      from sklearn.pipeline import make_pipeline
```

```
[In]: # target and split
      y, split = da.irr.get_task_target_and_split("regression")
      X_train, X_test = X[split != "test"], X[split == "test"]
      y_train, y_test = y[split != "test"], y[split == "test"]
      X_train.shape, y_train.shape, X_test.shape, y_test.shape
```

[Out]: ((18, 9, 59), (18,), (6, 9, 59), (6,))

```
[In]: from sklearn.dummy import DummyRegressor
```

```
[In]: from sktime.transformations.panel.reduce import Tabularizer

      dummy_pipeline = make_pipeline(
          Tabularizer(),
          DummyRegressor()
      )
      model = dummy_pipeline
      model.fit(X_train, y_train)
      mean_absolute_percentage_error(y_test, model.predict(X_test))
```

[Out]: 0.09721919149160385

```
[In]: from aeon.transformations.collection.dictionary_based import BORF
      from sklearn.linear_model import LassoCV

      borf_pipeline = make_pipeline(
          BORF(),
          LassoCV(),
      )
      model = borf_pipeline
      model.fit(X_train, y_train)
      mean_absolute_percentage_error(y_test, model.predict(X_test))
```

[Out]: 0.07554878610867717

# Notebook: Dataset Conversion

**The "Long Format"**

The basic format to convert any dataset to our representation is the long format. The long format is simply a tuple:

`(time_series_id, channel_id, timestamp, value, static_var_1, static_var_2, ...)`.

If your dataset contains rows that are in this format, you are almost good to go. Else, there will be a little bit of preprocessing to do.

**Case 1. (easy) Your dataset is already in the long format**

Let's assume for now your dataset is already in this form. Here is a minimal working example.

```python
[28]: import pandas as pd
      import numpy as np
```

```python
[29]: df = pd.DataFrame(
          {
              "time_series_id": np.random.choice(["A", "B", "C"], size=100),
              "channel_id": np.random.choice(["X", "Y", "Z"], size=100),
              "timestamp": pd.date_range("2023-01-01", periods=100, freq="H"),
              "value": np.random.randn(100),
          }
      )
      df["labels"] = df["time_series_id"].map(
          {"A": 0, "B": 1, "C": 1}
      )  # let's say we have labels
      df.head()
```

```
[29]:    time_series_id channel_id           timestamp     value  labels
      0               B          Y 2023-01-01 00:00:00  0.105162       1
      1               B          Z 2023-01-01 01:00:00 -0.573337       1
      2               B          X 2023-01-01 02:00:00 -1.973967       1
      3               C          Y 2023-01-01 03:00:00  0.656065       1
      4               A          Y 2023-01-01 04:00:00 -0.500246       0
```

```python
[30]: # Let's save this dataframe to a CSV file
      df.to_csv("your_original_dataset.csv", index=False)
```

```python
[31]: # the csv file can be converted to our format using our interface

      from pyrregular.io_utils import read_csv
      from pyrregular.reader_interface import ReaderInterface
      from pyrregular.accessor import IrregularAccessor

      class YourDataset(ReaderInterface):
```

```python
    @staticmethod
    def read_original_version(verbose=False):
        return read_csv(
            filenames="your_original_dataset.csv",
            ts_id="time_series_id",
            time_id="timestamp",
            signal_id="channel_id",
            value_id="value",
            dims={
                "ts_id": [
                    "labels"
                ], # static variable that depends on the time series id
                "signal_id": [],
                "time_id": [],
            },
            time_index_as_datetime=False,
            verbose=verbose,
        )
```

```python
[32]: da = YourDataset.read_original_version(True)
      da
```

Getting dataset metadata: 0it [00:00, ?it/s]

Reading dataset:    0%|              | 0/100 [00:00<?, ?it/s]

```
[32]: <xarray.DataArray (ts_id: 3, signal_id: 3, time_id: 100)> Size: 3kB
      <COO: shape=(3, 3, 100), dtype=float64, nnz=100, fill_value=nan>
      Coordinates:
        * time_id    (time_id) <U19 8kB '2023-01-01 00:00:00' ... '2023-01-05 03:00...
          labels     (ts_id) int64 24B 0 1 1
        * ts_id      (ts_id) <U1 12B 'A' 'B' 'C'
        * signal_id  (signal_id) <U1 12B 'X' 'Y' 'Z'
```

If you don't know if a variable is static, or to which dimension it depends from, you can check it.

```python
[33]: from pyrregular.data_utils import import infer_static_columns

      infer_static_columns(df, "time_series_id")
```

```
[33]: ['labels']
```

The dataset can be saved with our custom accessor

```python
[34]: da.irr.to_hdf5("your_dataset.h5")
```

And then loaded directly with xarray

```python
[35]: import xarray as xr
```

```
[36]:  da2 = xr.load_dataset("your_dataset.h5", engine="pyrregular")
       da2
```

```
[36]:  <xarray.Dataset> Size: 11kB
       Dimensions:     (ts_id: 3, signal_id: 3, time_id: 100)
       Coordinates:
           labels      (ts_id) int32 12B 0 1 1
         * signal_id   (signal_id) <U1 12B 'X' 'Y' 'Z'
         * time_id     (time_id) <U19 8kB '2023-01-01 00:00:00' ... '2023-01-05 03:00...
         * ts_id       (ts_id) <U1 12B 'A' 'B' 'C'
       Data variables:
           data        (ts_id, signal_id, time_id) float64 3kB <COO: nnz=100,
       fill_value=nan>
```

**Case 2. Your dataset is not in the long format**

Let's say you have a 3d numpy array, containing the time series, and a numpy array containing only the labels.

```
[37]:  import numpy as np

       shape = (10, 2, 100)  # 10 time series, 2 channels, 100 timestamps
       data = np.full(shape, np.nan)
       mask = np.random.rand(*shape) < 0.35
       data[mask] = np.random.randn(mask.sum())
       labels = np.random.randint(0, 2, shape[0])

       np.save("your_more_complex_dataset.npy", data)
       np.save("your_more_complex_dataset_labels.npy", labels)

       data.shape, labels.shape
```

```
[37]:  ((10, 2, 100), (10,))
```

You need only a function that takes the data and the labels, and returns a dataframe in the long format, yielding it row by row.

```
[38]:  def read_your_dataset(filenames):
           data = np.load(filenames["data"])
           labels = np.load(filenames["labels"])
           ts_ids, signal_ids, timestamps = np.indices(shape)
           ts_ids, signal_ids, timestamps = ts_ids.ravel(), signal_ids.ravel(),␣
       ↪timestamps.ravel()

           for ts_id, signal_id, timestamp in zip(ts_ids, signal_ids, timestamps):
               value = data[ts_id, signal_id, timestamp]
               if np.isnan(value):
                   continue
```

```
        label = labels[ts_id]
        yield dict(
            time_series_id=ts_id,
            channel_id=signal_id,
            timestamp=timestamp,
            value=value,
            labels=label,
        )
```

```
[39]:  from pyrregular.io_utils import read_csv
       from pyrregular.reader_interface import ReaderInterface
       from pyrregular.accessor import IrregularAccessor

       class YourDataset(ReaderInterface):
           @staticmethod
           def read_original_version(verbose=False):
               return read_csv(
                   filenames={
                       "data": "your_more_complex_dataset.npy",
                       "labels": "your_more_complex_dataset_labels.npy",
                   },
                   ts_id="time_series_id",
                   time_id="timestamp",
                   signal_id="channel_id",
                   value_id="value",
                   dims={
                       "ts_id": [
                           "labels"
                       ],  # static variable that depends on the time series id
                       "signal_id": [],
                       "time_id": [],
                   },
                   reader_fun=read_your_dataset,
                   time_index_as_datetime=False,
                   verbose=verbose,
                   attrs={
                       "authors": "Bond, James Bond",  # you can add any attribute you␣
       ↪want
                   }
               )
```

```
[40]:  da = YourDataset.read_original_version(True)
       da
```

```
Getting dataset metadata: 0it [00:00, ?it/s]
```

```
Reading dataset:   0%|          | 0/720 [00:00<?, ?it/s]
```

```
[40]: <xarray.DataArray (ts_id: 10, signal_id: 2, time_id: 100)> Size: 23kB
      <COO: shape=(10, 2, 100), dtype=float64, nnz=720, fill_value=nan>
      Coordinates:
        * time_id    (time_id) int64 800B 0 1 2 3 4 5 6 7 ... 92 93 94 95 96 97 98 99
          labels     (ts_id) int64 80B 0 0 0 1 1 1 0 1 1 0
        * ts_id      (ts_id) <U21 840B '0' '1' '2' '3' '4' '5' '6' '7' '8' '9'
        * signal_id  (signal_id) <U21 168B '0' '1'
      Attributes:
          authors:  Bond, James Bond
```