# OpenReview forum: "PYRREGULAR: A Unified Framework for Irregular Time Series, with Classification Benchmarks"
_ICLR.cc/2026/Conference — ICLR 2026 Poster_

### Official Review · Reviewer_r4xR · 2025-10-30

**Soundness:** 4
**Presentation:** 3
**Contribution:** 3
**Rating:** 6
**Confidence:** 4

**Summary:**

This paper introduces a unified preprocessing framework for irregular time series (ITS). The framework standardizes common steps and ships with curated datasets and baseline implementations. According to the paper, the pipeline supports multiple ITS tasks and provides a consistent and computationally efficient interface that lowers the barrier to reproducing baselines and evaluating new models across diverse benchmarks.

**Strengths:**

+ It handles a real problem in ITS learning research, despite not being a traditional ICLR paper
+ The framework covers multiple ITS settings, improving comparability across papers
+ The proposed pipeline to Raw ITS into model-ready tensors can accelerate experimentation and deployment

**Weaknesses:**

+ As the main contribution is the framework itself, the paper could be more focused on how it was implemented instead of evaluating the included models.
+ As it was proposed for unified ITS, some relevant families seem absent from the main implementation/benchmark (e.g., latent ODE[1]/RNN variants beyond NCDE, state-space models/SSMs, and foundation models as TabPFN[2]) despite being used for ITS with relevant results.
+ TIMESNET is reported as a transformer-based model, but it is not based on the attention mechanism.


[1] Rubanova, Yulia, Ricky TQ Chen, and David K. Duvenaud. "Latent ordinary differential equations for irregularly-sampled time series." Advances in neural information processing systems 32 (2019).

[2] Hollmann, Noah, et al. "Tabpfn: A transformer that solves small tabular classification problems in a second." arXiv preprint arXiv:2207.01848 (2022).

**Questions:**

+ As the MIMIC-III is available only to credentialed users, does the framework preprocessing code support the whole dataset?
+ What determined the models chosen for inclusion? Were there any barriers to adding existing models from other families?

**Details Of Ethics Concerns:**

.

---

> ### Author Response · Authors · 2025-11-14
>
> Thank you for your precise comments and for acknowledging the usefulness of "improving comparability across papers" and "accelerate experimentation and deployment".
> Below, we answer your questions. Modifications with respect to the original text are reported in blue in the revised manuscript.
>
> ### W1. As the main contribution is the framework itself, the paper could be more focused on how it was implemented instead of evaluating the included models.
>
> To be honest, we found it very hard to balance the framework and benchmark parts of this paper in a satisfactory way. On the one hand, we believe the framework in itself could be a contribution; however, this was not the opinion of several early reviewers. The framework description, while somewhat brief, gives an idea of how we tackled the issues of irregularity in Python, while the classification benchmark serves as proof that our framework really works and can be used to compare models from several libraries on a large collection of datasets. More practical details on implementation and usage can be found in Appendices F and G and, of course, in the included code. Extensive documentation also had to be removed due to anonymity concerns.
>
> ### W2/Q2.  What determined the models chosen for inclusion? Were there any barriers to adding existing models from other families? As it was proposed for unified ITS, some relevant families seem absent from the main implementation/benchmark (e.g., latent ODE[1]/RNN variants beyond NCDE, state-space models/SSMs, and foundation models as TabPFN[2]) despite being used for ITS with relevant results.
>
> This is a fair critique, and we openly acknowledge that the scope of this work is not to be fully comprehensive. As stated in the **Models** paragraph on page 6: "we limit our evaluation to classifiers that inherently support irregular inputs and are available in the aforementioned libraries" (sktime, aeon, tslearn, diffrax, and pypots). Unfortunately, these libraries do not yet include state-space models or foundation models for classification. We are working on adding them, as noted in Appendix F, but unlike the models already supported, their codebases are far less standardized and spread across multiple repositories. Incorporating a comprehensive set of models from these families would demand substantial effort, closer to what a dedicated benchmark paper would require.
>
> Regarding differential equation approaches, we did not include latent ODEs in the benchmark (but we already reference them in lines 38–39 in the Introduction), as, to the best of our knowledge, they are rarely used for classification, contrary to NCDEs, which were benchmarked on classification in the original paper [\*]. Regarding TabPFN, we were not aware that this was currently used on irregular time series, as, to our understanding, the main scope seemed to be small, regular tabular problems. Thank you for pointing this out; we included it as a reference for models that we will explore in the future (Appendix F).
>
> [\*] Kidger, Patrick, et al. "Neural controlled differential equations for irregular time series." (2020)
>
>
> ### W3. TIMESNET is reported as a transformer-based model, but it is not based on the attention mechanism.
>
> Thank you for catching that; indeed, it is more correct to refer to it as an inception-style model. We modified the text accordingly.
>
>
>
> ### Q1. As the MIMIC-III is available only to credentialed users, does the framework preprocessing code support the whole dataset?
>
> The version that we make directly available in our framework is the open-source demo version. As the real dataset follows the same standards, we do not foresee any issues in using the extended version in place of the demo.

---

### Official Review · Reviewer_CxTm · 2025-10-30

**Soundness:** 3
**Presentation:** 3
**Contribution:** 3
**Rating:** 6
**Confidence:** 4

**Summary:**

This paper introduces pyrregular, a unified framework designed to standardize the representation, analysis, and benchmarking of irregular time series (ITS) data. It proposes a clear taxonomy of irregularity types, implements an interoperable array structure that combines the flexibility of xarray with the memory efficiency of sparse COO tensors, and compiles a large repository of 34 naturally irregular datasets. The authors conduct a comprehensive evaluation of 12 classifiers spanning diverse modeling paradigms (including statistical, neural, and differential-equation-based approaches), offering the first systematic and reproducible benchmark for ITS classification. The work fills an important gap by centralizing disparate research efforts on irregular time series and providing a standardized, extensible foundation for future studies.

**Strengths:**

1. The paper clearly identifies a major gap in the field: the lack of interoperable tools and standardized benchmarks for irregular time series classification, which has long hindered cross-domain reproducibility and comparison.

2. The proposed array format elegantly combines xarray and sparse COO representations to achieve both flexibility and memory efficiency, while supporting multiple types of irregularity (uneven sampling, partial observation, raggedness).

3. The authors assemble a substantial suite of 34 real-world ITS datasets and evaluate 12 state-of-the-art models across multiple irregularity types, dataset scales, and sequence characteristics. The empirical analysis is thorough and yields valuable insights into the comparative strengths of classical versus deep models for ITS classification.

4. The framework, benchmark design, and dataset curation will likely become an important community resource for reproducible research and future extensions beyond classification.

**Weaknesses:**

1. While the benchmark covers a diverse set of classical and neural classifiers, recent developments in foundation or LLM-based time-series models (e.g., Time-LLM, CALF, or multimodal pretraining frameworks) are not discussed. Including such models, even conceptually, could contextualize where pyrregular fits within the broader trend toward generalist temporal modeling.

2. The paper briefly mentions runtime comparisons, but a deeper discussion of computational efficiency, scalability with data size, and memory footprint across model classes would strengthen the benchmarking narrative—especially given that one of pyrregular’s core motivations is interoperability and resource efficiency.

3. The paper focuses exclusively on classification, leaving forecasting, anomaly detection, and imputation for future work. A brief demonstration or pilot benchmark on another task could have further illustrated the generality of the proposed framework.

4.  For complex datasets such as MIMIC-III or PhysioNet 2019, a more in-depth analysis of model behavior and domain-specific challenges (e.g., handling clinical missingness patterns or label imbalance) would make the study more informative for practitioners in healthcare and other applied domains.

**Questions:**

see above

---

> ### Author Response · Authors · 2025-11-14
>
> Thank you for your comments and for agreeing on the relevance and importance of addressing this research gap, as well as the thoroughness of the experiments.
> Below, we answer your questions. Modifications with respect to the original text are reported in blue in the revised manuscript.
>
> ### W1. While the benchmark covers a diverse set of classical and neural classifiers, recent developments in foundation or LLM-based time-series models (e.g., Time-LLM, CALF, or multimodal pretraining frameworks) are not discussed. Including such models, even conceptually, could contextualize where pyrregular fits within the broader trend toward generalist temporal modelling.
>
> We acknowledge that model choice was dictated by practical considerations
> such as library availability and interface compatibility. As we state on page 6: "we limit our evaluation to classifiers that inherently support irregular inputs and are available in the aforementioned libraries" (sktime, aeon, tslearn, diffrax, and pypots). The approaches you mention (Time-LLM, CALF), while very relevant in general, are tailored for forecasting tasks, not classification. We agree that multimodal pretraining frameworks and time series foundation models are a rising topic in the literature; however, the main focus seems to be on regular time series.
> Based on your comment, we added a discussion about generalist temporal modelling, including the approaches you mentioned, to Appendix F, as well as in the conclusions, as future models to explore.
>
> ### W2. The paper briefly mentions runtime comparisons, but a deeper discussion of computational efficiency, scalability with data size, and memory footprint across model classes would strengthen the benchmarking narrative—especially given that one of pyrregular’s core motivations is interoperability and resource efficiency.
>
> Thank you, indeed R1 raised the same point. We already analyzed time and space complexity both empirically and theoretically in Appendix E of the submitted version for the biggest dataset in our repository, showing the efficiency of `pyrregular` in such cases. However, we did a poor job of referencing that in the main paper, so you might have missed it. We moved that appendix to a dedicated section in the main text of the revised version of the paper (Section 5.2: Complexity and Scalability).
>
> ### W3. The paper focuses exclusively on classification, leaving forecasting, anomaly detection, and imputation for future work. A brief demonstration or pilot benchmark on another task could have further illustrated the generality of the proposed framework.
>
> We fully agree on the relevance of such tasks. However, we are very open in the conclusion in saying that they are beyond the scope of this paper. That said, the paper already contained a brief discussion and technical details for such tasks in Appendix G (now Appendix F in the revision). Following your suggestion, we added a small, proof-of-concept regression benchmark on a few datasets and models (p. 40), as well as a code example for a regression task on the Garment dataset (p. 50). Note that the latter can also be seen as a point forecasting task, as we are predicting the future raw productivity.
>
>
> ### W4. For complex datasets such as MIMIC-III or PhysioNet 2019, a more in-depth analysis of model behavior and domain-specific challenges (e.g., handling clinical missingness patterns or label imbalance) would make the study more informative for practitioners in healthcare and other applied domains.
>
> We agree that deeper, dataset-specific analyses could benefit practitioners from different domains. However, our focus here is to provide a unified framework for working with irregular data and a general classification benchmark to demonstrate its usefulness. The paper is already dense and extensive, and we need to draw the line somewhere. We believe this work is a good starting point for research questions of this kind, and we have integrated your suggestions into the conclusions as future work.

---

### Official Review · Reviewer_rR2X · 2025-10-31

**Soundness:** 3
**Presentation:** 3
**Contribution:** 3
**Rating:** 6
**Confidence:** 4

**Summary:**

This paper studied the fragmentation in irregular time series research by proposing a unified framework and standardized dataset repository for irregular time series classification. Irregular time series, characterized by uneven sampling, partial observation, and raggedness, are common in many applications

**Strengths:**

The paper focuses on long-standing pain points in irregular time series research such as fragmented tools, lack of standardized benchmarks, and reliance on artificially induced irregularity.  The benchmark design is rigorous and comprehensive. Authors have evaluates a total of twelve methods over 34 datasets.

**Weaknesses:**

The paper studies exclusively on classification tasks and excludes other important time series tasks  such as forecasting, anomaly detection from benchmarking. While the paper mentions potential extensions, authors did not provide any technical details or preliminary results for these tasks.  Despite emphasizing practicality, authors provide incomplete details on computational ciost while it reports training/inference delay for classifiers, this paper did not analyze how the framework scales with increasing dataset.

**Questions:**

Can authors explicitly define how it distinguishes natural irregularity from artificial irregularity ?

---

> ### Author Response · Authors · 2025-11-14
>
> Thank you for acknowledging that "the benchmark design is rigorous and comprehensive". We truly appreciate your positive comment.
> Below, we answer your questions. Modifications with respect to the original text are reported in blue in the revised manuscript.
>
> ### Q1. Can authors explicitly define how it distinguishes natural irregularity from artificial irregularity?
>
> With artificial irregularity we refer to otherwise regular datasets in which missingness is injected (basically replacing observed data with nans).
> To make a clear example, taking the GunPoint dataset and replacing some values with nans is artificially induced missingness.
> This is a very common practice in the literature but inevitably alters the nature of the original data, as it overlooks structural missingness tied to data collection [\*]. In this work we use datasets in which the missingness was intrinsic, i.e., missing observations were truly missing. A brief discussion on this point is available in the first paragraphs of Appendix B.
>
> [\*] Mitra, Robin, et al. "Learning from data with structured missingness." Nature Machine Intelligence 5.1 (2023): 13-23.
>
>
> ### W1. The paper studies exclusively on classification tasks and excludes other important time series tasks such as forecasting, anomaly detection from benchmarking. While the paper mentions potential extensions, authors did not provide any technical details or preliminary results for these tasks.
>
> We fully agree on the relevance of such tasks, as we stated in our conclusion.
> We have made it very clear that they are beyond the scope of this paper, which we openly set to classification since the paper's title. This work is already dense and extensive, and we strongly believe that including such tasks in a satisfactory manner would require a full paper for each of them: we needed to draw a line somewhere.
>
> Appendix G of the submitted version (now Appendix F in the revision) already included some preliminary details for each of the future work tasks. To further prove the extensibility of `pyrregular`, we added to that appendix a proof of concept regression benchmark on a few datasets and models (p. 40), and also a code example (p. 50), including a working pipeline on a regression task on the Garment dataset. Note that the latter could also be viewed as a point-forecasting task, as the predicted raw productivity is in the future with respect to the training data.
>
> ### W2. Despite emphasizing practicality, authors provide incomplete details on computational cost while it reports training/inference delay for classifiers, this paper did not analyze how the framework scales with increasing dataset.
>
> We analyzed time and space complexity both empirically and theoretically in Appendix E of the submitted version for the biggest datasets in our repository, proving the efficiency of `pyrregular` in such cases. We believe you might have missed that, as we did a poor job of referencing it in the main paper. The same concern was also raised by R2, so we moved the appendix to a dedicated section in the main text (Section 5.2: Complexity and Scalability).

---

### Author Response · Authors · 2025-12-02
**Rebuttal Summary**

We want to thank all reviewers for their fair and on-point feedback, and for acknowledging the robustness and usefulness of the proposed framework and benchmark, as well as the relevance of the literature/tools gap tackled.
As the discussion with reviewers was unfortunately not possible, we summarize the common review points and the actions we took to address them in our rebuttal. Changes to the manuscript are highlighted in blue.

**Focus on classification.**
We better specified that the focus of the benchmarks is classification, as we openly stated in the paper's title and in the main text as well as in the appendices. We are aware and acknowledge that benchmarks on other tasks are very relevant for irregular time series, but introducing them in a systematic way would require an effort comparable to writing another full publication.
We also better stressed that the framework is **already capable** of supporting other tasks, e.g., time series **regression**, for which we introduced a **new pilot benchmark** in the rebuttal.

**Add efficiency benchmarks for the framework.**
These benchmarks were already present but not well referenced (so they may have been missed by reviewers). **We moved them to the main text in the rebuttal** (Section 5.2).

**Baseline methods.**
Reviewers asked the reason for not including more baselines. Some of the approaches suggested by the reviewers target forecasting or assume tabular inputs, and therefore cannot be applied directly to **irregular** time series **classification**. We clarified that our benchmark includes, at submission time, all 12 methods from five major time-series libraries that **natively support this task**. Additional baselines are planned for future work, as many candidate methods have scattered and non-standardized codebases that require substantial integration effort beyond the scope of the present study.

---

### Meta-Review · Area_Chair_nNm5 · 2026-01-12

**Summary:**

This paper introduces a unified framework and a standardized dataset repository for irregular time series classification. This paper provides a clear taxonomy of irregularity types, and a sizable repository of 34 naturally irregular datasets paired with a systematic benchmark of 12 representative methods. The resulting benchmark is rigorous and will likely become a useful community resource for reproducible comparison and faster experimentation. The main limitations are the initial focus on classification (with other tasks deferred) and originally under-emphasized efficiency/scalability reporting, as well as the absence of some model families (e.g., SSM/latent-ODE variants, foundation-model style baselines). However, the rebuttal clarifies the intended scope and provides a pilot benchmark beyond classification. Overall, I recommend the acceptance of this manuscript based on the rebuttal.

**Reviewer Concerns:**

The rebuttal clarifies the intended scope and provides a pilot benchmark beyond classification.

**Reviewer Scores:**

The reviewers are likely to maintain their positive scores on this paper.

---

### Decision · Program_Chairs · 2026-01-26

Accept (Poster)